# Exploring Specular Reflection Inconsistency for Generalizable Face Forgery Detection

**Hongyan Fei**[1,2]  **Zexi Jia**[3†]  **Chuanwei Huang**[1,2]  **Jinchao Zhang**[3]  **Jie Zhou**[3]

[1]School of Intelligence Science and Technology, Peking University
[2]State Key Laboratory of General Artificial Intelligence, Peking University
[3]WeChat AI, Tencent Inc
hongyanfei@stu.pku.edu.cn  huangcw@pku.edu.cn
{zexijia, dayerzhang, withtomzhou}@tencent.com
[†]Corresponding author.

## Abstract

Detecting deepfakes has become increasingly challenging as forgery faces synthesized by AI-generated methods, particularly diffusion models, achieve unprecedented quality and resolution. Existing forgery detection approaches relying on spatial and frequency features demonstrate limited efficacy against high-quality, entirely synthesized forgeries. In this paper, we propose a novel detection method grounded in the observation that facial attributes governed by complex physical laws and multiple parameters are inherently difficult to replicate. Specifically, we focus on illumination, particularly the specular reflection component in the Phong illumination model, which poses the greatest replication challenge due to its parametric complexity and nonlinear formulation. We introduce a fast and accurate face texture estimation method based on Retinex theory to enable precise specular reflection separation. Furthermore, drawing from the mathematical formulation of specular reflection, we posit that forgery evidence manifests not only in the specular reflection itself but also in its relationship with corresponding face texture and direct light. To address this issue, we design the Specular-Reflection-Inconsistency-Network (SRI-Net), incorporating a two-stage cross-attention mechanism to capture these correlations and integrate specular reflection related features with image features for robust forgery detection. Experimental results demonstrate that our method achieves superior performance on both traditional deepfake datasets and generative deepfake datasets, particularly those containing diffusion-generated forgery faces.

## 1 Introduction

With the rapid advancement of face manipulation techniques, generating highly realistic forgery face images and videos has become increasingly effortless (Rombach et al., 2022; Chen et al., 2025), and the potential abuse of this technology raises significant security concerns. Particularly with the rise of AI-generation tools such as diffusion models, which enable more sophisticated, realistic, and higher-resolution entire synthesized forgeries, making the challenge of face forgery detection becomes even harder (Kawar et al., 2023; Rosberg et al., 2023; Kim et al., 2025).

Current face forgery detection methods primarily identify forgery evidence by analyzing discriminative features extracted from the spatial (Li et al., 2020a; Cao et al., 2022) or frequency (Tan et al., 2024) domains of face images, as well as leveraging pre-trained features (Cui et al., 2025) from models such as CLIP (Radford et al., 2021). However, these approaches face significant limitations as texture inconsistencies in contemporary forgery faces, particularly those entirely synthesized by generative models, become increasingly subtle. Moreover, forgery evidence from different generation methods may manifest in distinct frequency bands, while pre-trained features lack domain-specific knowledge and interpretability. These factors collectively contribute to the declining detection performance of existing methods when confronted with sophisticated synthetic faces.

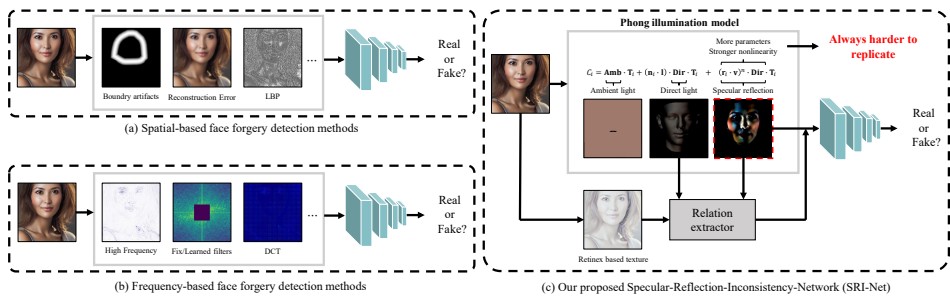

Figure 1: The visualization of (a) Spatial-based face forgery detection methods, (b) Frequency-based face forgery detection methods and (c) Our proposed Specular-Reflection-Inconsistency-Network (SRI-Net). SRI-Net analyzes that specular reflection is more difficult to replicate based on the mathematical form of the general Phong illumination model and contains generalizable forgery evidence.

In this paper, we propose an approach to localize forgery evidence from the perspective of face generation. Specifically, we posit that despite the remarkable realism achieved by state-of-the-art forgery techniques, they remain constrained by a fundamental principle: attributes governed by more estimated parameters and more complex physical laws are inherently more challenging to replicate accurately. Illumination represents a particularly complex physical phenomenon, as it is simultaneously governed by 2D local texture statistics, 3D global environmental conditions, and illumination model constraints, making it a promising avenue for investigation. As illustrated in Fig. 1 (c), under the Phong illumination model, face illumination comprises ambient light, direct light, and specular reflection. The mathematical representation of the specular reflection component involves more parameters and exhibits stronger nonlinearity, making it inherently more difficult to accurately replicate.

Based on this analysis, we hypothesize that specular reflection contains more generalizable forgery evidence. In this paper, we introduce the 3D Morphable Model (3DMM) (Blanz & Vetter, 2003) and computer graphics rendering algorithms to achieve illumination/texture separation and specular reflection extraction. However, since the widely used Basel Face Model (BFM) based texture estimation (Paysan et al., 2009) cannot capture fine-grained facial textures, the extracted specular reflection component inevitably contains erroneous details. To address this limitation, we propose a faster and more accurate face texture estimation method based on Retinex theory (Land & McCann, 1971) to replace the BFM model. Under the constraints of Retinex-based texture, we achieve more precise extraction of specular reflection. Furthermore, according to the analytical expression of specular reflection in the Phong illumination model, its estimation depends on both direct light intensity and reflective material properties. Since human faces share similar skin properties, the relative material characteristics can be approximated through face texture. Consequently, forgery evidence should manifest not only in the specular reflection component itself but also in its relationships with direct light and face texture. To exploit these relationships, we design the Specular-Reflection-Inconsistency-Network (SRI-Net) with a two-stage cross-attention structure to effectively capture the correlations among these attributes. The network subsequently integrates specular reflection related features with image features for final real/fake decision.

Our contributions can be summarized as follows:

1) We identify specular reflection as a robust forgery indicator due to its complex multi-parameter formulation, stronger nonlinearity and inherent difficulty to replicate accurately.

2) We introduce a Retinex-based face texture estimation method that enables faster and more precise specular reflection extraction.

3) We design SRI-Net with two-stage cross-attention to exploit the relationships among specular reflection, face texture, and direct light.

4) Extensive experiments demonstrate superior performance on both traditional and generative deepfake datasets.

## 2 RELATED WORK

### 2.1 GENERAL FACE FORGERY DETECTION

Many previous studies have focused on general face forgery detection. From the perspective of feature extraction, these approaches can be broadly categorized into spatial-domain and frequency-domain approaches. Spatial-domain methods primarily analyze pixel-level forgery evidence and inconsistencies in forgery images. MesoNet (Afchar et al., 2018) and Face X-ray (Li et al., 2020a) focused on mesoscopic properties or boundary discrepancies to distinguish manipulated faces. Some works (Wang et al., 2020; Wang & Deng, 2021; Cao et al., 2022) employed techniques such as adversarial training and reconstruction for enriching training data. Frequency-domain approaches (Masi et al., 2020; Qian et al., 2020; Jeong et al., 2022; Tan et al., 2024) decomposed images into frequency components using Fourier or wavelet transforms to detect forgery traces that are less perceptible in the spatial domain. From the perspective of learning generalized representations, SBI (Shiohara & Yamasaki, 2022) and SLADD (Chen et al., 2022b) explicitly minimized domain gaps to improve cross-dataset detection. Some works (Sun et al., 2020; Yan et al., 2024a) employed data augmentation and synthesis in latent space to boost model robustness. Some methods (Yan et al., 2023; Dong et al., 2023) incorporated generalized forgery cues through specialized learning frameworks for more discriminative detection. Some methods (Cui et al., 2025; Sun et al., 2025) leverage pretrained features from models such as CLIP. However, as forgery faces become increasingly realistic and forgery evidence is distributed unevenly across frequency bands, and with new forgery techniques continuously emerging, the effectiveness of existing detection methods declines.

### 2.2 ILLUMINATION-BASED DETECTION

Several existing methods (Peng et al., 2016; Zhu et al., 2021) detected forgery evidence by extracting illumination cues from 3D disentangled faces under the Lambertian assumption, demonstrating notable performance. Some works (Hu et al., 2021; Ebihara et al., 2020; Seibold et al., 2018) captured the local specular reflection on the cornea or nose for face forgery detection and face anti-spoofing. However, recent studies (Du et al., 2023; Zhan et al., 2024) revealed that diffusion models can effectively interpret geometric patterns and generate realistic shading estimations, thereby diminishing the reliability of such illumination-based detection approaches.

Nevertheless, the fundamental principle that attributes requiring more estimated parameters remain inherently more challenging to replicate remains unchanged. Given that illumination constitutes a complex physical process, it warrants further exploration. Therefore, we propose extending the Lambertian assumption to the more physically realistic Phong illumination model and precisely extract the complete and precise specular reflection component under 3D physical constraints, capturing the potential forgery evidence within it.

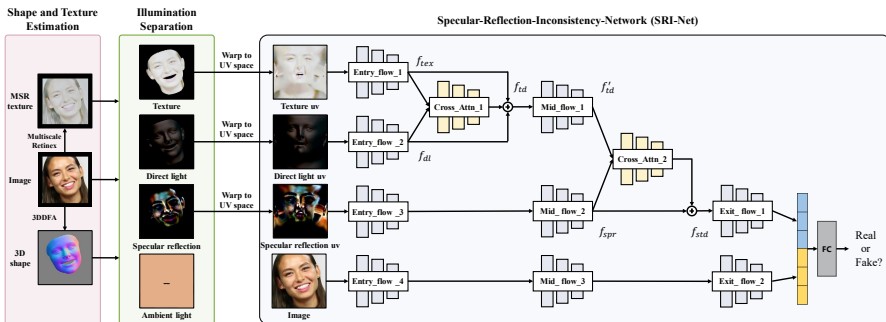

Figure 2: The framework of our proposed face forgery detection method. First, we use 3DDFA to extract 3D shape and propose a fast and accurate Retinex-based method for texture extraction. Next, we employ spherical harmonic model to fit ambient and direct light, extracting specular reflection through a residual based approach under Retinex-based texture constraints. We then propose the Specular-Reflection-Inconsistency-Network (SRI-Net) with a two-stage cross-attention structure to capture correlations among specular reflection, texture, and direct light. Finally, SRI-Net combines these specular reflection related features with image features for final real/fake decision.

## 3 METHOD

In this section, we first analyze face illumination under the Phong illumination model and demonstrate that specular reflection is more difficult to replicate and contains more generalizable forgery evidence. Subsequently, we propose a Retinex theory based face texture extraction method to achieve faster and more accurate specular reflection extraction. Finally, we present SRI-Net with a two-stage cross-attention architecture to capture the correlations of specular reflection, texture, and direct light, enabling more comprehensive forgery evidence extraction. The framework of our proposed method is illustrated in Fig. 2.

### 3.1 FACE ILLUMINATION ANALYSIS

From the perspective of face generation, a face image is composed of 3D structure and color information:

$$\mathbf{I}_{syn} = \mathcal{R}(\mathbf{S}, \mathbf{C}), \tag{1}$$

where $\mathcal{R}$ is the renderer, $\mathbf{S}$ is the 3D shape, and $\mathbf{C}$ represents the color information that combines texture (i.e., skin albedo) and illumination effects for each vertex on the 3D shape. The 3D shape is the easiest to replicate, as there is almost no occurrence of incompatible facial topology in existing forgery faces. While the color information $\mathbf{C}$ integrates both texture and illumination.

We posit that although state-of-the-art forgery techniques have achieved remarkable realism, they remain constrained by a fundamental principle in machine learning: highly complex and non-linear functions are inherently more difficult for models to learn accurately from finite data (Bishop & Nasrabadi, 2006). This is reflected in the well-known bias–variance trade-off, wherein approximating a complex target function demands substantial model capacity and data, often resulting in failure modes in the most challenging sub-components.

Based on this principle, the illumination component, as a complex physical process, is simultaneously constrained by 2D local texture statistics, 3D global environmental conditions, and illumination model constraints. Therefore, further investigation is warranted to extract potential forgery evidence from illumination-related aspects of color information.

The Phong illumination model is a widely used local illumination model that approximates the interaction of light with surfaces by combining ambient light, direct light and specular reflection components. Ambient light is uniform and illuminates all surfaces equally, which ensures that objects are visible even in areas not directly illuminated. Direct light is scattered across a surface based on its angle to the surface normal. Specular reflection creates shiny highlights, dependent on the surface's reflection of the light. Under Phong assumption, the RGB value for each vertex $C_i$ on $\mathbf{S}$ is computed as follows:

$$C_i = \mathbf{Amb} * \mathbf{T}_i + \langle \mathbf{n}_i, \mathbf{l} \rangle \cdot \mathbf{Dir} * \mathbf{T}_i + \langle \mathbf{r}_i, \mathbf{v} \rangle^n \cdot \mathbf{Dir} * \mathbf{T}_i, \tag{2}$$

where $\mathbf{Amb}$ is ambient light, $\mathbf{Dir}$ is direct light, $\mathbf{T}$ is the texture (i.e., skin albedo), $\langle \mathbf{r}_i, \mathbf{v} \rangle^n \cdot \mathbf{Dir}$ is specular reflection, $\langle, \rangle$ denotes the inner product operation, and $*$ denotes the element-wise multiplication. $\mathbf{n}_i$ is the vertex normal of the 3D shape, $\mathbf{l}$ is the direct light direction, $\mathbf{r}_i$ is the reflection direction which can be computed by $\mathbf{r}_i = 2\langle \mathbf{n}_i, \mathbf{l} \rangle \mathbf{n}_i - \mathbf{l}$, $\mathbf{v}$ is the viewer direction, and $n$ is the specular exponent of surface material. It can be observed that among the three types of illuminations, the mathematical formulation of specular reflection is more complex, involving six estimated parameters and a stronger nonlinear representation in exponential form. This makes it inherently more difficult to replicate.

Due to the fact that the Phong illumination model is a general model with clear physical significance, this characteristic where specular reflection is more difficult to replicate can be applied across various forgery methods, including current generative forgery techniques. Therefore, we posit that the specular reflection contains more generalizable forgery evidence.

### 3.2 SPECULAR REFLECTION EXTRACTION

To extract the specular reflection component, two main steps are involved according to Equ. 2: 1) illumination and texture separation, and 2) illumination components separation.

**Illumination and texture separation**

Existing methods typically perform illumination and texture separation using an analysis-by-synthesis approach within the 3D Morphable Model (3DMM) (Blanz & Vetter, 2003). The illumination and texture are parameterized through the spherical harmonic model (Zhang & Samaras, 2006) and the PCA texture model of Basel Face Model (BFM) (Paysan et al., 2009), with iterative optimization of the coefficients to obtain the illumination and texture components. In spherical harmonic model, face color under arbitrary illumination can be represented by the linear combination:

$$\mathbf{I(S)} = (\mathbf{H}\gamma) \cdot \mathbf{T}, \tag{3}$$

where $\mathbf{H} = [\mathbf{h}_1, \mathbf{h}_2, ..., \mathbf{h}_n]$ denotes a set of harmonic reflectance functions forming orthonormal bases to model illumination-induced brightness variations, while $\gamma = [\gamma_1, \gamma_2, ..., \gamma_n]$ represents the $n$-dimensional reflectance parameters requiring estimation. Notably, $\mathbf{h}_1$ is isotropic and $\mathbf{h}_1 \cdot \gamma_1$ can represent ambient light , whereas the directional components $[\mathbf{h}_2 \cdot \gamma_2, ..., \mathbf{h}_9 \cdot \gamma_9]$ can represent direct light. Basis with $n>9$ can be used to fit illumination with more detailed illumination, but the computational cost also increases significantly.

In PCA texture model of BFM, the texture is parameterized in the following form:

$$\mathbf{T} = \overline{\mathbf{T}}_{bfm} + \mathbf{B}\beta, \tag{4}$$

where $\overline{\mathbf{T}}_{bfm}$ is the mean face texture of BFM, $\mathbf{B}$ is the 199 dimensional principle axes, $\beta$ is the corresponding texture parameter. Based on the above equations, the parameters can be optimized through iterative methods by:

$$Argmin_{\gamma,\beta}\|\mathbf{I} - \mathbf{I}_{\text{syn}}(S, \gamma, \beta)\|. \tag{5}$$

where $\mathbf{I}$ is the original face image. The estimated $\mathbf{H}\gamma$ and $\overline{\mathbf{T}}_{bfm} + \mathbf{B}\beta$ can represent the separated illumination and texture.

However, the PCA bases $\mathbf{B}$ are obtained by eigendecomposition of the covariance matrix of 200 3D face scans in BFM, capturing dominant shape variations but failing to reconstruct identity fine-grained texture details due to dimensionality reduction and linear modeling constraints. Therefore, in real-world scenario, the face texture should be formulated as follows:

$$\mathbf{T} = \overline{\mathbf{T}}_{bfm} + \mathbf{B}\beta + \mathbf{T}_{id}, \tag{6}$$

where $\mathbf{T}_{id}$ is the identity texture details of corresponding face image. This indicates that the result derived from Equ. 5 contains a residual fine-grained texture detail $\mathbf{T}_{id}$ that the linear PCA model of the BFM fails to capture in comparison to the real texture $\mathbf{T}$.

Furthermore, inaccurate estimation of the texture will further lead to inaccurate estimation of the illumination. Particularly, due to the more complex and fine-grained nature of the specular reflection, it is more significantly affected when constrained by an inaccurate texture. Therefore, it is necessary to propose a new texture estimation method with a smaller residual to the real texture.

To address this issue, we propose a Retinex theory (Land & McCann, 1971) based texture extraction method for faster and more accurate estimation of specular reflection. Retinex theory is grounded in the physical model of image formation, which assumes that an observed image $I(x, y)$ is the product of illumination $L(x, y)$ and albedo $R(x, y)$, expressed as $I(x, y) = L(x, y) \cdot R(x, y)$.

To separate these components, Retinex employs a logarithmic transformation, converting the multiplicative relationship into an additive one:

$$\log I(x, y) = \log L(x, y) + \log R(x, y), \tag{7}$$

where the illumination $L(x, y)$ is typically estimated using low-pass filtering (e.g., Gaussian filtering) of the input image. This choice is theoretically justified by the assumption that illumination varies spatially smoothly, while albedo contains high-frequency details such as identity texture. By applying a Gaussian filter $G_\sigma$, the illumination is approximated as $G_\sigma(I(x, y))$, and the albedo can be obtained by:

$$\log R(x, y) = \log I(x, y) - \log[G_\sigma(I(x, y))], \tag{8}$$

where logarithmic form $\log R(x, y)$ is preferred over linear form $R(x, y)$ to represent the texture in Retinex theory. This is because it avoids nonlinear exponential transformations that could reintroduce illumination artifacts or cause numerical instability. The log-domain processing maintains

illumination invariance while better preserving fine surface details, making it more suitable for computational analysis of intrinsic surface properties under varying illumination conditions. Thus, the texture $\mathbf{T}_{re}$ of face image obtained by Retinex can be represented by the following formula:

$$\mathbf{T}_{re} = \log R. \tag{9}$$

Furthermore, to simultaneously address both global and local illumination variations, we employ Multi-Scale Retinex (MSR) to enhance robustness under complex illumination conditions. Specifically, large-scale Gaussian kernels are utilized to capture broad illumination gradients, while small-scale kernels supplement local illumination details, thereby enabling stable texture extraction across diverse illumination scenarios. The formula is presented as follows:

$$\mathbf{T}_{msr} = \frac{1}{N} \sum_{i=1}^{N} \left( \log I - \log \left[ G_{\sigma_i}(I) \right] \right), \tag{10}$$

where $G_{\sigma_i}$ denotes Gaussian kernels with varying scales $\sigma_i$, and $N$ represents the number of scales. This multi-scale integration ensures comprehensive illumination normalization while preserving fine-grained texture information. Thus, the illumination and texture separation process can be transformed from Equ. 5 into the following formula:

$$Argmin_{\gamma} \| \mathbf{I} - \mathbf{I}_{\text{syn}}(S, \gamma, \mathbf{T}_{msr}) \|. \tag{11}$$

Notably, substituting $\mathbf{T} = \overline{\mathbf{T}} + \mathbf{B}\beta$ with $\mathbf{T} = \mathbf{T}_{msr}$ not only makes the texture estimation more accurate but also eliminates the necessity to estimate the $\beta$ parameter. This approach circumvents the additional computational burden associated with the iterative optimization of both $\beta$ and $\gamma$ parameters required for the simultaneous optimization of illumination and texture. As a result, the processing time for single-image illumination and texture separation is reduced from 0.78s to 0.29s, significantly improving efficiency and enabling rapid large-scale data processing.

**Illumination components separation**

Based on Equ. 11, the spherical harmonic coefficients can be estimated. An intuitive approach is to utilize $\mathbf{h}_1 \cdot \gamma_1$ as the ambient light, $[\mathbf{h}_2 \cdot \gamma_2, ..., \mathbf{h}_9 \cdot \gamma_9]$ as direct light, and higher order basis along with their coefficients to represent the specular reflection. However, this approach faces similar issues to the PCA model of BFM, namely that the specular reflection fitted using spherical harmonic bases still exhibits residuals when compared to the real specular reflection. Additionally, the estimation of higher order spherical harmonic coefficients introduces significant computation cost.

To address this issue, we propose separating the specular reflection through a residual based approach. We utilize the first nine basis functions of the spherical harmonic model to capture coarse illumination variations, which effectively fit the ambient light and direct light while being insensitive to fine-grained specular reflection. Therefore, the separation of specular reflection can be achieved in the following form:

$$\mathbf{SPR} = (\mathbf{I} - (\mathbf{H}\gamma)_{(1-9)} \cdot \mathbf{T}_{msr}) / \mathbf{T}_{msr}, \tag{12}$$

where $\mathbf{SPR}$ is specular reflection. $(\mathbf{H}\gamma)_{(1-9)}$ represents the first nine spherical harmonic bases along with the coefficients estimated according to Equ. 11.

Fig. 3 (a) presents visualization of using (1) the mean texture $\overline{\mathbf{T}}_{bfm}$ and (2) the fitted texture $\overline{\mathbf{T}}_{bfm} + \mathbf{B}\beta$ from BFM and (3) the Retinex-based texture $\mathbf{T}_{msr}$ as texture $\mathbf{T}$, along with their corresponding obtained specular reflection, where the 3D shape is obtained by 3DDFA (Guo et al., 2020) in real-time. The results show that our proposed $\mathbf{T}_{msr}$ achieves more accurate texture estimation compared to $\overline{\mathbf{T}}_{bfm}$ and $\overline{\mathbf{T}}_{bfm} + \mathbf{B}\beta$, thereby enabling fine-grained removal of identity texture from specular reflection. Fig. 3 (b) presents a comparison of detailed differences in specular reflection. The red bounding boxes demonstrate that specular reflection with $\mathbf{T}_{msr}$ as constraint achieves more accurate removal of identity texture (e.g., in ocular and labial regions), while the green bounding boxes confirm that the extracted specular reflection better aligns with physical realism.

It is worth noting that while Retinex theory has been previously applied in face forensics tasks (Chen et al., 2019; 2022a), their objective was to leverage Retinex-based image enhancement to reveal subtle spatial-domain artifacts in forged regions, treating the Retinex output as a new input modality for the network. In our proposed method, the novelty lies not in the use of Retinex-based texture estimation itself, but in its application as a precision tool for 3D illumination decomposition. This

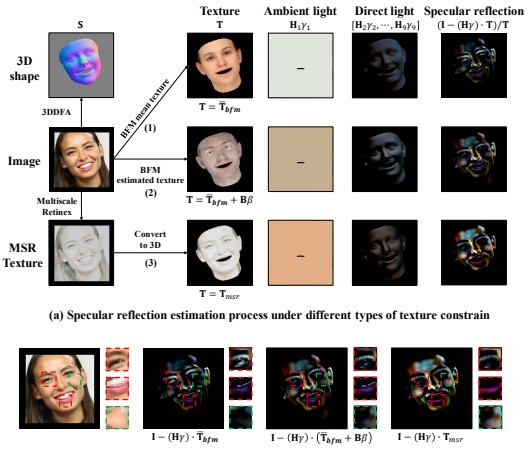

Figure 3: The visualization of (a) Specular reflection estimation process under different types of texture constraints and (b) Comparison of specular reflection detailed difference.

enables more accurate estimation of specular reflection, thereby capturing more robust and physically grounded forgery evidence.

### 3.3 SRI-NET

We propose a Specular-Reflection-Inconsistency-Network (SRI-Net) with Xception (Chollet, 2017) as backbone, with its architecture illustrated in Fig. 2. As shown in Equ. 2, the specular reflection intensity is determined by the light direction, 3D shape vertex normal, viewer direction, specular exponent of surface material, direct light intensity, and face texture. Therefore, forgery evidence should manifest not only in the specular reflection itself but also in its correlations with the aforementioned attributes. Given that human faces are all made of skin, the relative material properties can be approximated through face texture.

To capture these correlations, we first flatten the specular reflection, texture, and direct light components into UV space to normalize their directions. Subsequently, we employ a two-stage cross-attention mechanism with residual connections to extract their correlations. The first stage captures the relationship between texture and direct light. Let $f_{tex}$ and $f_{dl}$ denote the flattened feature maps of texture and direct light after the entry flow. The aggregated feature is computed as:

$$f_{td} = \text{Softmax}\left(\frac{f_{tex}f_{dl}^T}{\sqrt{d}}\right)f_{dl} + f_{tex} + f_{dl}, \tag{13}$$

where $d$ is the feature dimension, and $f_{td}$ encodes both features and their correlations. The second stage captures the correlations between specular reflection and the texture-direct light features. Let $f'_{td}$ be the processed $f_{td}$ through the middle flow, and $f_{spr}$ be the specular reflection features after passing through both entry and middle flows:

$$f_{std} = \text{Softmax}\left(\frac{f'_{td}f_{spr}^T}{\sqrt{d}}\right)f_{spr} + f_{spr}, \tag{14}$$

where $f_{std}$ aggregates the specular reflection features along with the correlations among specular reflection, texture, and direct light.

Meanwhile, since the original image retains the initial forgery evidence, we introduce an image branch to extract subtle information that may be distorted during specular reflection extraction, serving as a complementary component. Finally, we fuse the specular reflection related features with the image features to form the feature vector for final real/fake decision.

# 4 EXPERIMENTS

In this section, we provide a comprehensive evaluation of SRI-Net, covering various aspects such as datasets, detection performance comparisons, and ablation studies to demonstrate the effectiveness of SRI-Net.

## 4.1 DATASETS

**Traditional datasets.** We employ the FaceForensics++ (FF++) (Rossler et al., 2019), CelebDF_v1, CelebDF_v2 (Li et al., 2020b), and DeepfakeDetection (DFD) (Google AI, 2019) datasets for evaluation. To simulate real-world conditions, we utilize the FF++ c23 (light compression) version as the training set in subsequent experiments.

**Generative datasets.** We employ Diffusion Facial Forgery (DiFF) (Cheng et al., 2024a) and DF40 (Yan et al., 2024b) datasets for evaluation. DiFF dataset contains over 500,000 images synthesized using various generation methods (e.g., Stable Diffusion XL (SDXL), Low-Rank Adaptation (LoRA)) across four categories: Text-to-Image (T2I), Image-to-Image (I2I), Face Swapping (FS) and Face Editing (FE). Additionally, DF40 dataset contains forged data generated within the FF++ and CelebDF domains to ensure method diversity while maintaining consistent data distribution.

**Evaluation Metrics.** We utilize the Frame-Level Area Under Curve (AUC) metric on generative deepfake datasets, and utilize both frame-level and video-level AUC on traditional deepfake datasets.

## 4.2 COMPARISON ON TRADITIONAL DATASETS

Table 1: Frame-level AUC (%) performance comparison on traditional Datasets.

| Method | Venue | Traditional Datasets | | | |
|---|---|---|---|---|---|
| | | CDF-v1 | CDF-v2 | DFD | Avg |
| FWA (Li & Lyu, 2018) | CVPRW'18 | 79.0 | 66.8 | 74.0 | 73.3 |
| CapsuleNet (Nguyen et al., 2019) | ICASSP'19 | 79.1 | 74.7 | 68.4 | 74.1 |
| CNN-Aug (Haliassos et al., 2022) | CVPR'20 | 74.2 | 70.3 | 64.6 | 69.7 |
| Face X-ray (Li et al., 2020a) | CVPR'20 | 70.9 | 67.9 | 76.6 | 71.8 |
| FFD (Dang et al., 2020) | CVPR'20 | 78.4 | 74.4 | 80.2 | 77.7 |
| F3Net (Qian et al., 2020) | ECCV'20 | 77.7 | 79.8 | 70.2 | 75.9 |
| SPSL (Liu et al., 2021) | CVPR'20 | 81.5 | 76.5 | 81.2 | 79.7 |
| SRM (Luo et al., 2021) | CVPR'21 | 79.3 | 75.5 | 81.2 | 78.7 |
| CORE (Ni et al., 2022) | CVPRW'22 | 78.0 | 74.3 | 80.2 | 77.5 |
| RECCE (Cao et al., 2022) | CVPR'22 | 76.8 | 73.2 | 81.2 | 77.1 |
| SBI (Shao et al., 2022) | CVPR'22 | - | 81.3 | 77.4 | - |
| UCF (Yan et al., 2023) | ICCV'23 | 77.9 | 75.3 | 80.7 | 78.0 |
| ED (Ba et al., 2024) | AAAI'24 | 81.8 | 86.4 | - | - |
| ProDet (Cheng et al., 2024b) | NIPS'24 | 90.9 | 84.2 | 84.8 | 86.6 |
| LSDA (Yan et al., 2024a) | CVPR'24 | 86.7 | 83.0 | 88.0 | 85.9 |
| Fada (Cui et al., 2025) | CVPR'25 | - | 83.7 | - | - |
| FIA-USA (Ma et al., 2025) | ARXIV'25 | 90.1 | 86.7 | 82.1 | 86.3 |
| SRI-Net (Ours) | - | **91.3** | **87.5** | **89.3** | **89.4** |

Table 2: Video-level AUC(%) performance comparison on traditional Datasets.

| Method | Traditional Datasets | |
|---|---|---|
| | CDF-v2 | DFD |
| Xception | 81.6 | 89.6 |
| EffiNet-B4 | 80.8 | 86.2 |
| RECCE | 82.3 | 89.1 |
| F3-Net | 78.9 | 84.4 |
| SBI | 90.6 | 88.2 |
| UCF | 83.7 | 86.7 |
| FIA-USA | 94.1 | - |
| SRI-Net (Ours) | **95.5** | **93.1** |

Tab. 1 and Tab. 2 present the frame-level and video-level AUC comparisons on traditional datasets, where video-level scores are computed by averaging the frame-level scores across all frames. SRI-

Net achieves state-of-the-art performance under both evaluation protocols. Specifically, it attains the highest AUC scores across all three datasets at the frame level, demonstrating its effectiveness in capturing subtle forgery evidence through explicit modeling of correlations among specular reflection, texture, and direct light. At the video level, SRI-Net maintains its superior performance despite relying solely on frame-wise score aggregation without exploiting temporal dependencies.

Table 3: Frame-level AUC (%) performance comparison on DF40 Dataset.

| Method | Venue | DF40 Dataset | | | | | | |
|---|---|---|---|---|---|---|---|---|
| | | uniface | e4s | facedancer | fsgan | inswap | simswap | Avg |
| RECCE | CVPR'22 | 84.2 | 65.2 | 78.3 | 88.4 | 79.5 | 73.0 | 78.1 |
| SBI | CVPR'22 | 64.4 | 69.0 | 44.7 | 87.9 | 63.3 | 56.8 | 64.4 |
| CORE | CVPRW'22 | 81.7 | 63.4 | 71.7 | 91.1 | 79.4 | 69.3 | 76.1 |
| IID | CVPR'23 | 79.5 | 71.0 | 79.0 | 86.4 | 74.4 | 64.0 | 75.7 |
| UCF | ICCV'23 | 78.7 | 69.2 | 80.0 | 88.1 | 76.8 | 64.9 | 76.3 |
| LSDA | CVPR'24 | 85.4 | 68.4 | 75.9 | 83.2 | 81.0 | 72.7 | 77.8 |
| CDFA | ECCV'24 | 76.5 | 67.4 | 75.4 | 84.8 | 72.0 | 76.1 | 75.4 |
| ProgressiveDet | NIPS'24 | 84.5 | 71.0 | 73.6 | 86.5 | 78.8 | 77.8 | 78.7 |
| FIA-USA | ARXIV'25 | 91.8 | 87.5 | 83.0 | 86.3 | 87.4 | **91.0** | 87.8 |
| SRI-Net (Ours) | - | **92.0** | **89.4** | **95.3** | **94.2** | **91.1** | 83.3 | **90.9** |

## 4.3 COMPARISON ON GENERATIVE DATASETS

Tab. 3 and Tab. 4 present frame-level AUC comparisons on the DF40 and DiFF datasets. For the six representative faceswap subsets in DF40 dataset, SRI-Net achieves a substantial improvement in average AUC from 87.8% to 90.9%. Similarly, on DiFF dataset, our method consistently outperforms existing approaches across all four subsets.

Notably, while conventional methods often experience performance degradation on generative datasets due to the reduced texture inconsistencies, SRI-Net maintains robust detection performance. This resilience can be attributed to the fact that specular reflection is inherently difficult to replicate accurately, and modeling their correlations with multiple facial attributes based on well established physical principles.

Table 4: Frame-level AUC (%) performance comparison on DiFF Dataset.

| Method | DiFF Dataset | | | | |
|---|---|---|---|---|---|
| | T2I | I2I | FS | FE | Avg |
| Xception | 62.4 | 56.8 | 86.0 | 58.6 | 66.0 |
| EffiNet-B4 | 74.1 | 57.3 | 82.1 | 57.2 | 67.6 |
| F3-Net | 66.9 | 67.7 | 81.0 | 60.6 | 69.1 |
| SBI | 80.2 | 80.4 | 85.1 | 68.8 | 78.6 |
| FIA-USA | 86.1 | 85.0 | 89.4 | 72.7 | 83.3 |
| SRI-Net (Ours) | **88.7** | **85.5** | **92.2** | **80.8** | **86.8** |

## 4.4 ABLATION STUDIES

In this section, we conduct ablation studies to analyze the SRI-Net, the experimental results are presented in Tab. 5.

**Effectiveness of each branch.** Rows 1-5 in Tab. 5 evaluate individual branch contributions. 'Img', 'SPR', and 'MSR' denote the use of the original image, specular reflection, and MSR-based texture as the sole input to XceptionNet. 'Img + SPR' refers to using both the original image and specular reflection as inputs, with their feature vectors concatenated as the output. Meanwhile, 'Img + SPR + MSR/Dir' corresponds to the SRI-Net architecture. It can be observed that SPR alone achieves considerable AUC performance. This strongly indicates that the specular reflection can effectively capture intrinsic forgery traces from images, which are discriminative in nature and demonstrate strong generalization across datasets. The combination of Img and SPR branches achieves superior performance, as the original image preserves undistorted forgery evidence while the SPR branch isolates difficult to replicate specular reflection that contains generalizable forgery evidence. MSR alone yields suboptimal performance. However, a marked improvement is observed when MSR is utilized in conjunction with the SPR and Dir (compare rows (4) and (5)). This contrast substantiates that the primary role of the MSR texture in our framework is not as a standalone feature, but as a key element in extracting the physically meaningful specular reflection inconsistency.

Table 5: Ablation study results on AUC (%) of SRI-Net.

| Method Settings | Datasets | | |
|---|---|---|---|
| | Celeb_v2 | DF40 | DiFF |
| (1) Img | 72.1 | 74.2 | 66.0 |
| (2) SPR | 75.7 | 78.8 | 73.9 |
| (3) MSR | 68.5 | 71.0 | 60.8 |
| (4) Img + SPR | 84.5 | 87.2 | 83.9 |
| (5) Img + SPR + MSR/Dir | 87.5 | 90.9 | 86.8 |
| (6) $\mathbf{T} = \overline{\mathbf{T}}_{bfm}$ | 83.3 | 85.4 | 79.9 |
| (7) $\mathbf{T} = \overline{\mathbf{T}}_{bfm} + \mathbf{B}\beta$ | 84.1 | 88.2 | 82.5 |
| (8) $\mathbf{T} = \mathbf{T}_{msr}$ | 87.5 | 90.9 | 86.8 |
| (9) w/o shape norm | 82.1 | 83.7 | 81.0 |
| (10) w/ shape norm | 87.5 | 90.9 | 86.8 |
| (11) Original Image | 87.5 | 90.9 | 86.8 |
| (12) Gaussian Blurring | 86.6 | 88.7 | 84.2 |
| (13) JPEG Compression | 87.1 | 90.2 | 86.4 |
| (14) Gaussian Noise | 87.0 | 88.9 | 85.5 |
| (15) Cross-Attention | 87.5 | 90.9 | 86.8 |
| (16) SE-block | 86.9 | 88.6 | 84.5 |
| (17) Concatenated | 85.6 | 88.1 | 83.7 |

**Effectiveness of Retinex-based Texture Extraction.** Rows 6-8 compare three texture extraction methods (Fig. 3). The Retinex-based $\mathbf{T}_{msr}$ achieves optimal performance by capturing fine-grained facial textures, enabling more precise separation of illumination components.

**Effectiveness of shape norm.** Rows 8-9 demonstrate that shape normalization improves performance by standardizing illumination directions, surface normals, and viewing directions, thereby enhancing the correlation capture between specular reflection, texture, and direct light components under the Phong illumination model.

**Robustness to common post-processing.** Rows 11-14 evaluate the robustness of our proposed SRI-Net under three types of common post-processing on test images: Gaussian Blurring, JPEG Compression, and Gaussian Noise. The results show that SRI-Net has demonstrated good stability, maintaining a high AUC performance. This stability stems from its physics-driven design: while such distortions often degrade high-frequency forgery evidence used by spatial methods, SRI-Net targets deeper physical inconsistencies in specular reflection, direct light and texture. These violations of illumination physics remain detectable even after image-level corruptions, ensuring more resilient forgery detection.

**The choice of attention mechanism.** Rows 15-17 compare fusion strategies for capturing inconsistencies among specular reflection, direct light, and texture. Cross-attention outperforms both simple feature concatenation and channel-weighting methods like the SE-block. We attribute this to its ability to dynamically compute weighted interactions across all spatial locations, which allows the model to discover and emphasize subtle, non-local inconsistencies between modalities. This provides a more expressive way to model their complex relationships compared to static or channel-level fusion operations.

## 5 CONCLUSION

In this paper, we build upon the principle that face attributes requiring more estimated parameters and more complex physical constraints are inherently harder to replicate accurately, and demonstrate that the specular reflection component in the Phong illumination model contains generalizable forgery evidence. To enable accurate illumination separation, we develop a faster and more accurate Retinex-based texture extraction method that facilitates precise specular reflection estimation. We then propose SRI-Net to capture the correlations between specular reflection, face texture, and direct light components, thereby enabling the extraction of more complete forgery evidence. Overall, we propose a novel approach for detecting forgery evidence under physically realistic constraints, demonstrating superior performance on both traditional deepfake datasets and more challenging generative deepfake datasets.

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

# A  APPENDIX

## A.1  THE USE OF LLM

We used Large Language Models (LLMs) to assist with writing polishing and writing refinement during the preparation of this manuscript. The LLMs were employed solely for improving grammar, sentence structure, and overall readability. All research ideas, data analysis, conclusions, and substantive content remain entirely the original work of us. We take full responsibility for the accuracy and integrity of all content presented in this paper.

## A.2  IMPLEMENT DETAILS

For training specifications, we employ Xception as the backbone, optimized with the Adam optimizer at an initial learning rate of 1e-3 for 80 epochs. During training on the FF++ dataset, we uniformly sample 32 frames per video sequence, and the input size is 256x256. Data augmentation strategies include HorizontalFlip, RandomCutout and AddGaussianNoise. The multiscale Retinex-based texture extraction method utilizes $\sigma$ values of [15, 80, 120] for face texture extraction.

## A.3  EXAMPLES OF SPECULAR REFLECTION EXTRACTION

In this section, we present additional samples of specular reflection extraction in Fig. 4. We utilize 3DDFA for 3D shape extraction and propose a fast and accurate Retinex-based method for texture extraction. Subsequently, we employ spherical harmonic model to fit ambient light and direct light, extracting specular reflection through a residual-based approach under Retinex-based texture constraints.

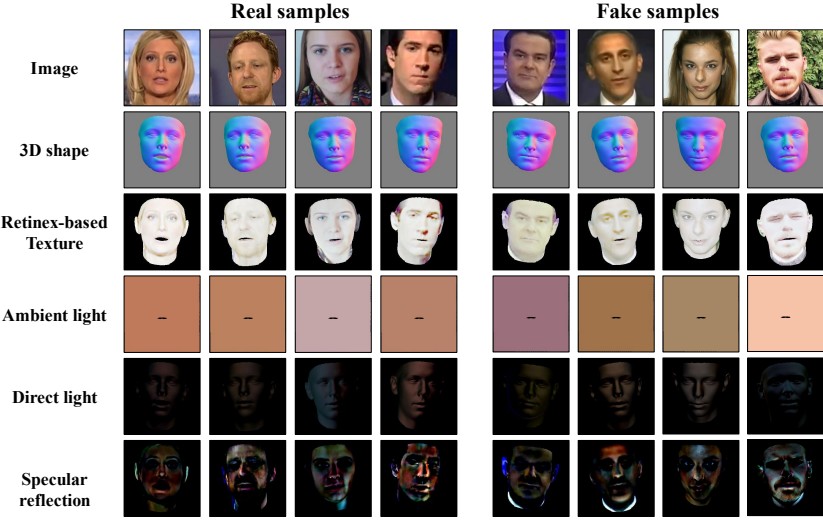

Figure 4: The visualization of Specular Reflection Extraction. The face image can be decomposed into 3D shape, Retinex-based texture, ambient light, direct light, and specular reflection under Phong illumination model constraints. The samples on the left are real samples, while the samples on the right are fake samples.

## A.4  ROBUSTNESS ACROSS SKIN TONES

This section investigates the impact of different skin tones on the performance of SRI-Net. The evaluation is conducted on the AI-Face dataset (Lin et al., 2025), which provides demographic attribute labels. The experimental results, summarized in Tab. 6, are reported for the test set stratified into three subsets: Light, Medium, and Dark, representing varying levels of skin tone. The results demonstrate that our method achieves superior and consistent performance across all three skin tone groups. This provides empirical evidence of its robustness to skin tones.

The stability can be explained through the Phong illumination model. Variations in skin tone primarily manifest as differences in the ambient light component of the model (as shown in Fig. 4), as this component interacts directly and uniformly with the skin's diffuse albedo. A key strength of our pipeline is its explicit separation of illumination. The Retinex-based texture extraction and subsequent spherical harmonic modeling collaboratively work to isolate and filter out ambient light component. By effectively removing the primary carrier of skin tone information from the analysis, our method minimizes its sensitivity to such variations. The subsequent specular reflection analysis is therefore conducted on a more normalized representation, leading to consistent performance across demographics. This design inherently decouples the detection from skin tone. We have added this analysis and the experimental results to the revised manuscript.

Table 6: AUC(%) performance comparison on AI-Face dataset.

| Method | Skin Tone | | |
|---|---|---|---|
| | Light | Medium | Dark |
| Xception (Chollet, 2017) | 97.69 | 98.44 | 98.88 |
| EffiNet-B4 (Tan & Le, 2019) | **99.23** | 98.94 | 97.59 |
| F3-Net (Qian et al., 2020) | 98.51 | 98.68 | 98.79 |
| CORE (Ni et al., 2022) | 97.80 | 98.47 | 98.79 |
| DAG-FDD (Ju et al., 2024) | 97.56 | 98.79 | 98.73 |
| SRI-Net (Ours) | 99.12 | **99.01** | **99.18** |

## A.5 LIMITATION ANALYSIS AND FUTURE WORK

Fig. 5 presents cases of misclassification. It can be observed that these cases are characterized by extreme facial poses or significant self-occlusion on the surface, resulting in inaccurate 3D shape fitting and consequently imprecise specular reflection extraction.

To address this issue, our planned future work will focus on two key technical directions to enhance the robustness of the 3D shape estimation:

The first approach is to leverage recent advances in highly robust 2D facial landmark detection that achieve pixel-level accuracy in unconstrained environments (e.g., Pixel-in-Pixel Net (Jin et al., 2021)) to correct inaccuracies in the initial 3D shape. The core idea is to compute the displacement vectors between the projected vertices of the estimated 3D shape and the corresponding pixel-accurate 2D landmarks. These vertex-wise offsets can then be interpolated and applied to refine the entire 3D shape, effectively pulling it into better alignment with the image evidence, even under challenging poses. Based on these offsets, we can correct deviations in the 3D shape caused by extreme facial poses or occlusions.

The second approach involves adopting more advanced 3D shape estimation frameworks, such as 3DDFA-V3 (Wang et al., 2024), which utilizes segmentation information for face reconstruction. This provides a more robust optimization objective than mere photometric alignment, leading to more stable and accurate 3D shapes in the presence of occlusions and extreme viewing angles.

## A.6 STATISTICAL VISUALIZATION ANALYSIS OF SPECULAR REFLECTION

In this section, we conduct both PCA visualization and score distribution analysis on the features extracted when using specular reflection as the sole input to XceptionNet (Row 2 of Tab. 5), evaluated on the DiFF dataset. The results are shown in Fig. 6 and Fig. 7. The PCA visualization, which projects the original 2048-dimensional feature vectors onto a 2D space, does not clearly reveal distinct distribution patterns between real and fake faces. This limited visual separability is expected given the model's AUC of 73.9% and the substantial information loss inherent in reducing high-dimensional features to just two dimensions. However, our score distribution histogram demonstrates a statistically significant divergence between real and fake samples. The clear separation in score distributions provides compelling evidence that the specular reflection component captures discriminative forgery traces capable of effectively distinguishing between real and fake faces.

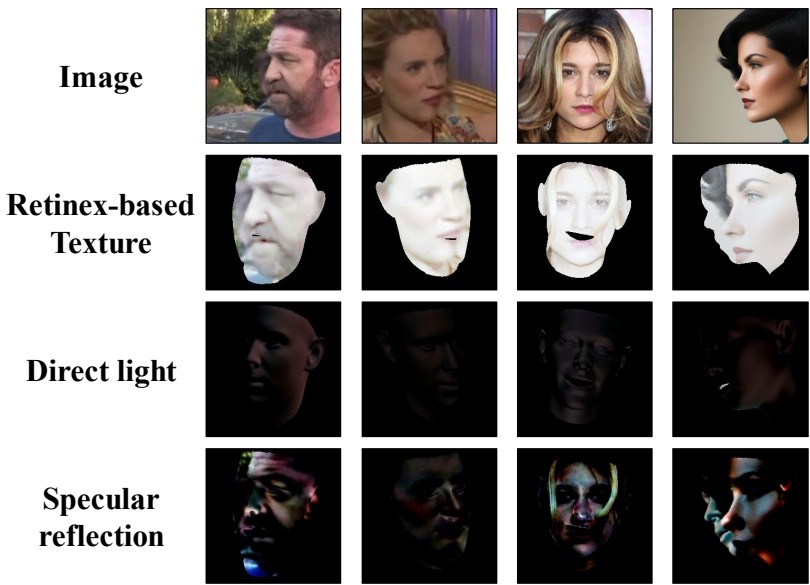

Figure 5: The visualization of misclassified cases. These cases are characterized by extreme facial poses or severe self-occlusion of the facial surface.

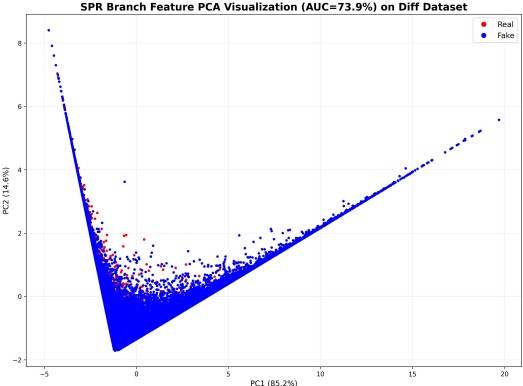

Figure 6: SPR branch feature PCA visualization.

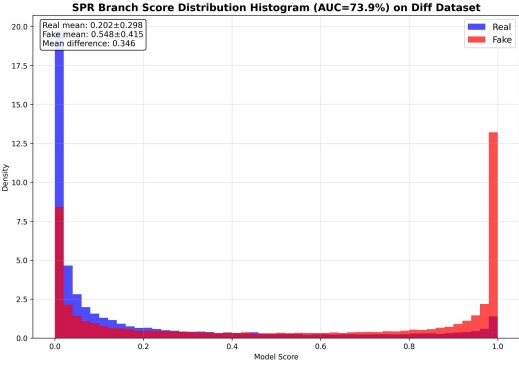

Figure 7: SPR branch score distribution histogram.

