# OpenReview forum: "Exploring Specular Reflection Inconsistency for Generalizable Face Forgery Detection"
_ICLR.cc/2026/Conference — ICLR 2026 Poster_

### Official Review · Reviewer_TtYa · 2025-10-26

**Soundness:** 2
**Presentation:** 2
**Contribution:** 2
**Rating:** 6
**Confidence:** 4

**Summary:**

In this paper, the authors propose the hypothesis that specular reflection, a phenomenon governed by complex physical laws, is difficult for AI models to simulate, and therefore can serve as a key cue for generalizable deepfake detection. Specifically, they first employ Retinex for fast and accurate extraction of facial textures, then achieve improved specular reflection separation under the Phong illumination model. Building on these steps, they introduce the Specular-Reflection-Inconsistency Network (SRI-Net), which leverages a two-stage cross-attention mechanism to integrate relationships among specular reflection, facial texture, and direct lighting. Extensive experiments on both traditional and newly released generative deepfake datasets demonstrate that the proposed method outperforms existing approaches.

**Strengths:**

1. The paper focuses on the physical properties of light–surface interaction, the proposed approach demonstrates improved robustness and universality.

2. The authors replace the traditional BFM PCA-based texture estimation with multi-scale texture extraction based on Retinex theory, and design SRI-Net to effectively model the relationships among direct illumination, texture, and specular reflection.

3. The method is thoroughly evaluated on multiple classic and state-of-the-art datasets, achieving significant performance gains on both frame-level and video-level metrics.

**Weaknesses:**

1. The central claim, i.e., “Specular reflection is hard to replicate”, is intuitive but not particularly novel. The paper does not provide a new theoretical explanation or quantitative physical analysis to substantiate why specular reflection is inherently more discriminative than other illumination features.

2. Although the authors aim to capture the inconsistencies among specular reflection, texture, and direct illumination, they do not provide any mathematical formulation or quantitative metric defining how such inconsistencies are measured.

3. Although the authors claim that specular reflection is difficult to counterfeit, their method depends on 3D shape estimation (via 3DDFA) and Retinex-based texture extraction, both of which may fail under unconstrained facial poses, occlusions, or complex lighting conditions. This limitation is only briefly mentioned in the appendix and lacks in-depth discussion or empirical validation under “in-the-wild” scenarios.

4. The rationale behind the two-stage structure of SRI-Net and the choice of attention mechanism as the optimal modeling strategy is insufficiently explained and would benefit from clearer justification.

**Questions:**

Please see weaknesses.

Overall, the reviewer thinks that this paper presents an interesting and promising perspective on leveraging physical properties of specular reflection for deepfake detection. However, the work would benefit from more comprehensive theoretical analysis to substantiate its central hypothesis and further strengthen the proposed framework.

---

> ### Author Response · Authors · 2025-11-19
> **Response to Reviewer TtYa (Part1)**
>
> Thank you for your thoughtful feedback and valuable suggestions, which helped us a lot to make the paper better. The responses to the concerns are as follows. We have revised the manuscript accordingly, and the changes have been highlighted in blue for your convenience.
>
> **1. Theoretical Foundation of the Central Claim**
> >W1: The central claim, i.e., “Specular reflection is hard to replicate”, is intuitive but not particularly novel. The paper does not provide a new theoretical explanation or quantitative physical analysis to substantiate why specular reflection is inherently more discriminative than other illumination features.
>
> Thank you for this insightful comment, which highlights the need to better articulate the theoretical foundation of our core premise. Our motivation stems from a fundamental principle in machine learning: highly complex and non-linear functions are inherently more difficult for models to learn accurately from finite data [1]. This is reflected in the well-known bias-variance trade-off, where approximating a complex target requires immense capacity and data, often leading to failure modes in the most challenging sub-components.
>
> We apply this principle to face forgery detection through the lens of physical image formation. The Formula of Phong illumination model provides a clear hierarchy of complexity: specular reflection is objectively more complex than ambient light and direct light due to its dependence on multiple parameters (viewer direction, surface normals, material exponent) and strong non-linearity (exponential term), making specular reflection more difficult for generators to replicate accurately. They must not only replicate texture but also the intricate, physics-grounded relationships governing light interaction. Thus, inaccuracies in replicating this complex phenomenon serve as a generalizable forgery evidence.
>
> * The formula of Phong illumination model:
> $C_i = {Amb} \ast {T}\_i + \langle {n}\_i , {l} \rangle \cdot {Dir} \ast {T}\_i + \langle{r}\_i , {v}\rangle^n \cdot{Dir} \ast {T}\_i$
>
> where $\mathbf{Amb}$ is ambient light, $\mathbf{Dir}$ is direct light, $\mathbf{T}$ is the texture (i.e., skin albedo), $\langle\mathbf{r}_i , \mathbf{v}\rangle^n \cdot\mathbf{Dir}$ is specular reflection, $\langle , \rangle$ denotes the inner product operation, and $\ast$ denotes the element-wise multiplication. $\mathbf{n}_i$ is the vertex normal of the 3D shape, $\mathbf{l}$ is the direct light direction, $\mathbf{r}_i$ is the reflection direction which can be computed by $\mathbf{r}_i = 2\langle\mathbf{n}_i , \mathbf{l}\rangle\mathbf{n}_i - \mathbf{l}$, $\mathbf{v}$ is the viewer direction, and $n$ is the specular exponent of surface material.
>
>
> Our proposed method is designed to explicitly detect these subtle inconsistencies. The superior performance of SRI-Net across diverse datasets provides strong empirical validation for this theoretically grounded approach. While the visual differences in Fig.3 may be subtle, as the artifacts are often embedded in complex high-frequency patterns. The primary purpose of Fig.3 is to demonstrate that a more accurate specular reflection is obtained when the Retinex-based texture is used as a constraint.
>
>
> [1] Pattern recognition and machine learning[M]. Springer, 2006.

---

> ### Author Response · Authors · 2025-11-19
> **Response to Reviewer TtYa (Part2)**
>
> **2. Measurement of Inconsistencies**
> >W2: Although the authors aim to capture the inconsistencies among specular reflection, texture, and direct illumination, they do not provide any mathematical formulation or quantitative metric defining how such inconsistencies are measured.
>
> Thank you for this insightful comment, which pushes us to clarify a key aspect of our methodology.
>
> The comment is correct that we do not provide an **explicit, hand-crafted mathematical formula or metric** to quantify the inconsistency. This is a deliberate design choice. The "inconsistencies" we refer to are **high-dimensional, non-linear, and relational patterns** among specular reflection, texture, and direct light. Manually defining a closed-form metric to capture such complex, emergent artifacts is extremely challenging.
>
> Instead of a pre-defined metric, our proposed two-stage cross-attention mechanism (Formula 13-14) serves as a learnable function that is optimized to discover and quantify these inconsistencies directly from data. The core of our approach is to allow the model to learn *how* these components should consistently relate under the Phong illumination model constraints, and to detect deviations from this learned norm.
>
> *   The first cross-attention stage (Formula 13) learns the baseline relationship between texture and direct light ($f_{td}$).
> *   The second stage (Formula 14) then evaluates how well the actual specular reflection aligns with this expected baseline. The output feature $f_{std}$ is, effectively, a **learned embedding that encodes the degree and nature of the multi-attribute inconsistency**.
>
> In other words, the "measurement" is performed implicitly but effectively by the cross-attention operations, which compute weighted interactions across all spatial locations and feature dimensions. The superior performance of SRI-Net across diverse datasets (Tabs. 1-4) provides strong empirical validation that this learned approach successfully captures a generalizable and discriminative signal of physical inconsistency.
>
> **3. 3D Shape Estimation Limitations and Mitigation Strategies**
> >W3: Although the authors claim that specular reflection is difficult to counterfeit, their method depends on 3D shape estimation (via 3DDFA) and Retinex-based texture extraction, both of which may fail under unconstrained facial poses, occlusions, or complex lighting conditions. This limitation is only briefly mentioned in the appendix and lacks in-depth discussion or empirical validation under “in-the-wild” scenarios.
>
> Thank you for this insightful comment. In the revised manuscript, we have expanded the discussion in Appendix A.5 to provide deeper insights and concrete strategies for mitigating failures caused by extreme poses and occlusions. Our planned future work will focus on two key technical directions to enhance the robustness of the 3D shape estimation:
>
> The first approach is to leverage recent advances in highly robust 2D facial landmark detection that achieve pixel-level accuracy in unconstrained environments (e.g., Pixel-in-Pixel Net [1]) to correct inaccuracies in the initial 3D shape. The core idea is to compute the displacement vectors between the projected vertices of the estimated 3D shape and the corresponding pixel-accurate 2D landmarks. These vertex-wise offsets can then be interpolated and applied to refine the entire 3D shape, effectively pulling it into better alignment with the image evidence, even under challenging poses. Based on these offsets, we can correct deviations in the 3D shape caused by extreme facial poses or occlusions.
>
> The second approach involves adopting more advanced 3D shape estimation frameworks, such as 3DDFA-V3 [2], which utilizes segmentation information for face reconstruction. This provides a more robust optimization objective than mere photometric alignment, leading to more stable and accurate 3D shapes in the presence of occlusions and extreme viewing angles.
>
> [1] Pixel-in-pixel Net: Towards Efficient Facial Landmark Detection in the wild. IJCV 2021.
>
> [2] 3D Face Reconstruction with the Geometric Guidance of Facial Part Segmentation. CVPR 2024.
>
> We have incorporated the above discussion into Appendix A.5 of the revised manuscript.

---

> ### Author Response · Authors · 2025-11-19
> **Response to Reviewer TtYa (Part3)**
>
> **4. Rationale for SRI-Net Architecture**
> >W4: The rationale behind the two-stage structure of SRI-Net and the choice of attention mechanism as the optimal modeling strategy is insufficiently explained and would benefit from clearer justification.
>
> Thank you for this question, which allows us to clarify the design rationale for SRI-Net.
>
> The two-stage structure is directly motivated by our physics-driven approach. The Phong illumination model implies that forgery evidence lies in the relationships between attributes. Therefore, we first model the fundamental interaction between texture and direct light in Stage 1. Stage 2 then evaluates how the observed specular reflection aligns with or deviates from this baseline, thereby explicitly capturing the complex, triple-attribute inconsistency that is difficult for generators to replicate.
>
> Regarding the choice of the attention mechanism, we have incorporated an ablation study in the revised manuscript to validate its optimality. The results confirm that the cross-attention strategy outperforms both simple feature concatenation and channel-weighting methods like the SE-block. We posit this is because the attention mechanism dynamically computes weighted interactions across all spatial locations, allowing it to discover and emphasize subtle, non-local inconsistencies between modalities, which is a more powerful way to model their relationships than a channel level fusion operation.
>
> **Ablation study regarding the choice of the attention mechanism.**
>
> | Method Settings   | Celeb_v2 | DF40 | DiFF |
> | :---------------- | :------- | :--- | :--- |
> | Cross-Attention   | **87.5**     | **90.9** | **86.8** |
> | SE-block          | 86.9     | 88.6 | 84.5 |
> | Concatenated      | 85.6     | 88.1 | 83.7 |

---

> > ### Comment · Reviewer_TtYa · 2025-11-24
> >
> > Thanks for the reply. Most of my concerns have been resolved. But I am still concerned that if there is no precise physical definition for the specular reflection inconsistency, then the method in this paper is actually more like intuitive data augmentation. That is, it has not changed the traditional idea of current deepfake detection. How do the authors precisely define the core contributions and insights of their method?

---

> > > ### Author Response · Authors · 2025-11-24
> > > **Response to Reviewer TtYa**
> > >
> > > Thank you for the follow-up question, which helps us clarify a potential misunderstanding and further elaborate on the physical grounding of our work. We apologize that our initial response may have given the impression that "inconsistency" lacks a physical definition. On the contrary, it is rigorously defined by the Phong illumination model.
> > >
> > > The physical definition of specular reflection inconsistency is rooted in the formula of the Phong illumination model:
> > >
> > > $C_i = \mathbf{Amb} \ast \mathbf{T}_i + \langle \mathbf{n}_i, \mathbf{l} \rangle \cdot \mathbf{Dir} \ast \mathbf{T}_i + \langle \mathbf{r}_i, \mathbf{v} \rangle^n \cdot \mathbf{Dir} \ast \mathbf{T}_i$.
> > >
> > > where $\mathbf{Amb}$ is ambient light, $\mathbf{Dir}$ is direct light, $\mathbf{T}$ is the texture (i.e., skin albedo), $\langle\mathbf{r}_i , \mathbf{v}\rangle^n \cdot\mathbf{Dir}$ is specular reflection, $\langle , \rangle$ denotes the inner product operation, and $\ast$ denotes the element-wise multiplication. $\mathbf{n}_i$ is the vertex normal of the 3D shape, $\mathbf{l}$ is the direct light direction, $\mathbf{r}_i$ is the reflection direction which can be computed by $\mathbf{r}_i = 2\langle\mathbf{n}_i , \mathbf{l}\rangle\mathbf{n}_i - \mathbf{l}$, $\mathbf{v}$ is the viewer direction, and $n$ is the specular exponent of surface material.
> > >
> > > The specular reflection component, $\mathbf{SPR} = \langle \mathbf{r}_i, \mathbf{v} \rangle^n \cdot \mathbf{Dir}$, establishes a precise physical constraint: for a given direct light direction $\mathbf{l}$, 3D shape vertex normal $\mathbf{n}_i$, viewer direction $\mathbf{v}$, direct light intensity $\mathbf{Dir}$, and face texture $\mathbf{T}_i$ (Given that human faces share similar skin properties, the relative material characteristics $n$ can be approximated through face texture.), the specular reflection must follow a specific, deterministic pattern in a real face.
> > >
> > > In our pipeline, we first flatten the specular reflection, face texture, and direct light components into UV space to normalize their directions (norm direct light direction $\mathbf{l}$, 3D shape vertex normal $\mathbf{n}_i$, viewer direction $\mathbf{v}$). In a real face, under the constraints of the Phong illumination model, the specular reflection exhibits a specific and consistent pattern relative to the corresponding face texture and direct light. However, as described in our response to W1, the specular reflection component is objectively more complex and difficult for generative models to replicate perfectly due to its dependence on multiple parameters and strong non-linearity. Therefore, in forged images, the specular reflection will deviate from the pattern characteristic of a real face given the corresponding texture and direct light. This deviation is the "specular reflection inconsistency" we defined, a breakdown in the multi-attribute relational constraint imposed by the Phong illumination model.
> > >
> > > Thus, the core of SRI-Net is not to learn an arbitrary pattern from the fusion of input components but to learn how these components should consistently relate under the Phong illumination model's constraints.
> > >
> > > *   The first cross-attention stage (Formula 13) learns the baseline relationship between texture and direct light ($f_{td}$).
> > > *   The second stage (Formula 14) then evaluates how well the actual specular reflection aligns with this expected baseline. The output feature $f_{std}$ is the learned embedding that encodes the degree and nature of the multi-attribute inconsistency under the Phong illumination model.
> > >
> > > In summary, our method is far from intuitive data augmentation; it is an approach that leverages a clear physical definition of relational constraints and employs a specialized network architecture to detect violations of these constraints, which are fundamental signatures of forgery evidence.

---

### Official Review · Reviewer_KKiD · 2025-10-30

**Soundness:** 3
**Presentation:** 3
**Contribution:** 3
**Rating:** 6
**Confidence:** 3

**Summary:**

The paper proposes a novel approach for face forgery detection by leveraging the inherent difficulty of replicating specular reflection under the Phong illumination model. The authors introduce a faster and more accurate Retinex-based texture extraction method to enhance the separation of illumination components, particularly specular reflection. They design the Specular-Reflection-Inconsistency-Network (SRI-Net), which uses a two-stage cross-attention mechanism to capture correlations between specular reflection, face texture, and direct light. Experimental results demonstrate that the method achieves state-of-the-art performance on both traditional deepfake datasets and generative datasets, showcasing its robustness and generalizability.

**Strengths:**

- Novelty

The paper introduces a unique perspective by focusing on specular reflection as a forgery indicator, which is grounded in well-established physical principles. This approach provides a fresh direction in deepfake detection research.
The use of the Phong illumination model and Retinex theory for texture extraction is innovative and enhances the accuracy of specular reflection separation.


- Strong Experimental Results

The method achieves superior frame-level and video-level AUC scores across traditional and generative datasets, outperforming existing state-of-the-art methods.
The robustness of the approach on high-quality generative datasets highlights its generalizability and resilience against subtle forgery evidence.

- Efficiency

The Retinex-based texture extraction method significantly reduces processing time compared to traditional iterative optimization methods, making it suitable for large-scale data processing.

**Weaknesses:**

- Limited Analysis of Failure Cases

While the paper briefly discusses misclassification due to extreme facial poses or occlusion, it does not provide detailed insights into how these issues could be mitigated or addressed in future work.

- Dependence on 3D Shape Fitting

The method relies heavily on accurate 3D shape fitting to extract specular reflection. In scenarios with severe occlusion or extreme poses, the performance may degrade significantly. The paper does not explore alternative solutions or enhancements for these situations.

- Complexity of Implementation

Although the method is efficient in terms of processing time, the overall pipeline, including Retinex-based texture extraction, spherical harmonic modeling, and SRI-Net architecture, might be challenging to implement and deploy in real-world applications.

- Limited Real-World Validation

The datasets used for evaluation, while diverse, may not fully represent the complexities of real-world deepfake scenarios. Additional experiments on real-world datasets or adversarially generated deepfakes could strengthen the paper's claims.

**Questions:**

see above

---

> ### Author Response · Authors · 2025-11-19
> **Response to Reviewer KKiD (Part1)**
>
> Thank you for your thoughtful feedback and valuable suggestions, which helped us a lot to make the paper better. The responses to the concerns are as follows. We have revised the manuscript accordingly, and the changes have been highlighted in blue for your convenience.
>
> **1. Mitigation Strategies for Failure Cases**
>
> >W1:(Limited Analysis of Failure Cases) While the paper briefly discusses misclassification due to extreme facial poses or occlusion, it does not provide detailed insights into how these issues could be mitigated or addressed in future work.
>
> Thank you for this insightful comment. In the revised manuscript, we have expanded the discussion in Appendix A.5 to provide deeper insights and concrete strategies for mitigating failures caused by extreme poses and occlusions. Our planned future work will focus on two key technical directions to enhance the robustness of the 3D shape estimation:
>
> The first approach is to leverage recent advances in highly robust 2D facial landmark detection that achieve pixel-level accuracy in unconstrained environments (e.g., Pixel-in-Pixel Net [1]) to correct inaccuracies in the initial 3D shape. The core idea is to compute the displacement vectors between the projected vertices of the estimated 3D shape and the corresponding pixel-accurate 2D landmarks. These vertex-wise offsets can then be interpolated and applied to refine the entire 3D shape, effectively pulling it into better alignment with the image evidence, even under challenging poses. Based on these offsets, we can correct deviations in the 3D shape caused by extreme facial poses or occlusions.
>
> The second approach involves adopting more advanced 3D shape estimation frameworks, such as 3DDFA-V3 [2], which utilizes segmentation information for face reconstruction. This provides a more robust optimization objective than mere photometric alignment, leading to more stable and accurate 3D shapes in the presence of occlusions and extreme viewing angles.
>
> [1] Pixel-in-pixel Net: Towards Efficient Facial Landmark Detection in the wild. IJCV 2021.
>
> [2] 3D Face Reconstruction with the Geometric Guidance of Facial Part Segmentation. CVPR 2024.
>
> We have incorporated the above discussion into Appendix A.5 of the revised manuscript.
>
> **2. Dependence on 3D Shape Fitting**
> >W2: (Dependence on 3D Shape Fitting) The method relies heavily on accurate 3D shape fitting to extract specular reflection. In scenarios with severe occlusion or extreme poses, the performance may degrade significantly. The paper does not explore alternative solutions or enhancements for these situations.
>
> Thank you for this valuable feedback. As detailed in our response to the previous comment W1, we have substantially expanded the analysis of failure cases and mitigation strategies in Appendix A.5 of the revised manuscript.
>
> Specifically, we discuss two concrete technical pathways for future work to enhance robustness:
>
> 1. Refining the initial 3D shape estimation by leveraging pixel-level accurate 2D landmarks to correct deviations caused by poses and occlusions.
>
> 2. Integrating more advanced 3D face reconstruction frameworks that utilize stronger geometric constraints, such as facial part segmentation.
>
> We believe that these proposed directions provide a clear and feasible roadmap to directly address the limitations of extreme poses and occlusions, thereby strengthening the practical applicability of our method.

---

> ### Author Response · Authors · 2025-11-19
> **Response to Reviewer KKiD (Part2)**
>
> **3. Complexity of Implementation**
> >W3: (Complexity of Implementation) Although the method is efficient in terms of processing time, the overall pipeline, including Retinex-based texture extraction, spherical harmonic modeling, and SRI-Net architecture, might be challenging to implement and deploy in real-world applications.
>
> Thank you for this practical observation. We acknowledge that our multi-stage pipeline introduces implementation complexity. However, this design is instrumental in achieving the state-of-the-art generalization performance demonstrated across diverse datasets. The complexity is mitigated by the pipeline's modularity and the efficiency of each component:
>
> 1. Retinex-based Texture Extraction: This is a classical and highly efficient image processing operation. Our multi-scale implementation is non-iterative and can be optimized for real-time execution.
> 2. Spherical Harmonic illumination Modeling & 3D Shape Fitting: We rely on 3DDFA for 3D shape estimation, which is a fast, pre-trained model. The Spherical Harmonic illumination estimation is a lightweight linear least-squares problem once the shape and texture are fixed. The replacement of the iterative BFM texture fitting with our Retinex method (as detailed in Sec. 3.2) reduced the single-image processing time from 0.78s to 0.29s, demonstrating a significant simplification and speed-up of this very step.
> 3. SRI-Net: This is a standard deep learning module that can be exported and deployed using high-performance inference engines (e.g., TensorRT, ONNX Runtime).
>
> For real-world deployment, the computationally heavy parts (3D shape fitting, texture extraction and illumination estimation) can be pre-computed offline, requiring only the lightweight SRI-Net forward pass for online detection. We believe the substantial gain in robustness against sophisticated forgeries justifies this structured, physics-based approach.
>
> **4. Real-World Validation**
> >W4 : (Limited Real-World Validation) The datasets used for evaluation, while diverse, may not fully represent the complexities of real-world deepfake scenarios. Additional experiments on real-world datasets or adversarially generated deepfakes could strengthen the paper's claims.
>
> Thank you for this valuable suggestion regarding real-world validation. We agree that robustness in complex, real-world scenarios is crucial.
>
> In our initial manuscript, we already evaluated our method on generative datasets (DF40 and DiFF) that contain forgeries from GAN and Diffusion models, which represent a significant step beyond traditional benchmarks. To directly and more comprehensively address the concern, we have now incorporated an additional evaluation on the large-scale, demographically diverse AI-Face dataset [1] in our revised manuscript.
>
> This dataset is particularly relevant for real-world validation as it simultaneously incorporates two critical dimensions of complexity: 1>A broad spectrum of demographic attributes (e.g., skin tone, age, gender), ensuring robustness across diverse populations. 2>A wide variety of modern generative methods. The results in Tab. 6 of the revised manuscript demonstrate that our method achieves superior and consistent performance across all three skin tone groups. This provides empirical evidence of its robustness, providing additional evidence for its generalizability and practical applicability in real-world settings.
>
> **AUC(%) performance comparison on AI-Face dataset.**
>
> | Method                      | Light | Medium | Dark |
> | :-------------------------- | :---- | :----- | :--- |
> | Xception                | 97.69 | 98.44  | 98.88 |
> | EffiNet-B4              | **99.23** | 98.94  | 97.59 |
> | F3-Net                  | 98.51 | 98.68  | 98.79 |
> | CORE                    | 97.80 | 98.47  | 98.79 |
> | DAG-FDD                 | 97.56 | 98.79  | 98.73 |
> | SRI-Net (Ours)          | 99.12 | **99.01** | **99.18** |
>
>
> [1] AI-Face: A Million-Scale Demographically Annotated AI-Generated Face Dataset and Fairness Benchmark. CVPR 2025.

---

> > ### Comment · Reviewer_KKiD · 2025-11-26
> > **keep the initial score**
> >
> > Thank you for your feedback. Most of the raised concerns have been adequately addressed, and thus I will maintain my original rating.

---

> > > ### Author Response · Authors · 2025-11-26
> > > **Response to Reviewer KKiD**
> > >
> > > Thank you for your kind assessment. We are pleased to know that our revisions successfully addressed your concerns.

---

### Official Review · Reviewer_yPYK · 2025-10-31

**Soundness:** 2
**Presentation:** 3
**Contribution:** 2
**Rating:** 4
**Confidence:** 4

**Summary:**

This manuscript presents a novel methodology for generalizable face forgery detection, underpinned by the observation that facial attributes governed by complex physical laws, particularly the specular reflection component, are inherently challenging for contemporary synthesis models to accurately replicate. The proposed approach systematically decomposes the facial image into fundamental physical components, including 3D shape, texture, and illumination, with the primary objective of isolating the specular reflection component for subsequent inconsistency analysis and detection. The core decomposition strategy employed is based on Retinex theory.

**Strengths:**

The manuscript is clearly written and easy to follow, with a concise method formulation and a framework that is simple yet demonstrably effective. The method achieves consistently strong results across a broad spectrum of face-swapping techniques, including both diffusion-based and GAN-based pipelines, suggesting good generalization beyond a single generator family. The ablation studies provide compelling evidence that each branch, as well as the method’s design/strategy choices, contributes positively to the overall performance.

**Weaknesses:**

1.In Section 3.2, the choice to employ Retinex theory and Multi-Scale Retinex (MSR) to achieve a 'smaller to the real texture' and enhance robustness is a major component of the methodology. However, these Retinex-based enhancement techniques are well-established, and their application alone does not appear to be a novel contribution. More critically, previous work [1], has utilized similar Retinex-based methods for image enhancement, yet this paper is not cited.

2.The work is motivated by the claim that the decomposed space is more complex and thereby facilitates distinguishing real from fake; however, no statistical or theoretical evidence is provided, and the visual comparisons in Figs. 3 and 4 fail to substantiate this.

[1] Chen, Han, Yuzhen Lin, and Bin Li. "Exposing face forgery clues via retinex-based image enhancement." Proceedings of the Asian Conference on Computer Vision. 2022.

**Questions:**

1.Could you elaborate on what is meant by '3D global environmental conditions' (line 176, Section 3.1). It is vague.

2.In Section 3.1 (line 185), there is a potential source of confusion regarding the notation in Formula (1) and Formula (2). Specifically, Formula (1) treats $C$ and $S$ as distinct variables, yet in Formula (2), $S$ is defined as being composed of elements $C_i$. The variable $C_i$ could easily be misinterpreted as a component of $C$, leading to ambiguity and obscuring the intended relationship between $C$, $S$, and $C_i$.

3.For the title 'Illumination and texture separation' in Section 3.2 (line 205), is the intention to decouple the texture and illumination components that were implicitly combined within the variable $C$ as introduced in Formula(1)?

4.The text spanning lines 236 to 247 in Section 3.2 should be broken down into smaller, logical paragraphs. The current dense formatting is difficult to read.
Regarding Formula (6), is there a specific model or methodology suggested for obtaining the identity transformation matrix $T_{id}$? For example, could a pre-trained model such as ArcFace be employed for this purpose, or is an alternative approach required?

---

> ### Author Response · Authors · 2025-11-19
> **Response to Reviewer yPYK (Part1)**
>
> Thank you for your thoughtful feedback and valuable suggestions, which helped us a lot to make the paper better. The responses to the concerns are as follows. We have revised the manuscript accordingly, and the changes have been highlighted in blue for your convenience.
>
> **1. The usage of Multi-Scale Retinex**
> >W1: In Section 3.2, the choice to employ Retinex theory and Multi-Scale Retinex (MSR) to achieve a 'smaller to the real texture' and enhance robustness is a major component of the methodology. However, these Retinex-based enhancement techniques are well-established, and their application alone does not appear to be a novel contribution. More critically, previous work [1], has utilized similar Retinex-based methods for image enhancement, yet this paper is not cited.
> [1] Chen, Han, Yuzhen Lin, and Bin Li. "Exposing face forgery clues via retinex-based image enhancement." Proceedings of the Asian Conference on Computer Vision. 2022.
>
> Thanks for your comment and for pointing out these relevant works. We agree that Retinex theory has been previously applied in face forensics tasks [1,2], and we have cited these papers in the revised manuscript to better situate our contribution. In our manuscript, the novelty lies not in the use of Retinex-based texture estimation, but in its novel role within our designed physics-driven framework for exposing forgery evidence.
>
> The objective of [1,2] is to exploit Retinex-based image enhancement to reveal subtle spatial domain artifacts in forged regions. In these approaches, Retinex representations serve as a new modality that is directly fed into the network as input.
>
> In contrast, our objective centers on physics-based illumination decomposition. Motivated by the Phong illumination model formulation, we identify specular reflection as a robust forgery indicator due to its complex multi-parameter formulation, stronger nonlinearity and inherent difficulty to replicate accurately. Our goal is to precisely isolate and extract the specular component, subsequently exploiting inconsistencies between specular reflection, face texture and direct light to capture forgery evidence. This represents a physics-grounded analytical framework rather than mere image enhancement.
>
> In our pipeline, Multi-Scale Retinex (MSR) serves not as an input modality but rather as a critical intermediate module that facilitates more accurate specular reflection extraction. As detailed in Section 3.2, the widely used Basel Face Model (BFM) fails to capture fine-grained facial textures, leading to contaminated specular reflection estimates. Our MSR-based texture $T_{msr}$ is specifically designed to replace and outperform the BFM texture model. This allows for a purer separation of the specular component via our proposed residual-based approach (Formula 12: ${SPR} = ({I} - ({H}\gamma)\_{(1-9)} \cdot {T}\_{msr})/{T}\_{msr}$).
>
> In general, while we build upon the established Retinex theory, its application in our work is contextually and functionally distinct. Previous works use it for general enhancement; we use it as a precision tool for 3D illumination decomposition to estimate more accurate specular reflection to capture more robust forgery evidence.
>
> [1] Attention-based Two-stream Convolutional Networks for Face Spoofing Detection. IEEE TIFS 2020.
>
> [2] Exposing Face Forgery Clues via Retinex-based Image Enhancement. ACCV 2022.

---

> ### Author Response · Authors · 2025-11-19
> **Response to Reviewer yPYK (Part2)**
>
> **2. Motivation and Theoretical Foundation**
> >W2: The work is motivated by the claim that the decomposed space is more complex and thereby facilitates distinguishing real from fake; however, no statistical or theoretical evidence is provided, and the visual comparisons in Figs. 3 and 4 fail to substantiate this.
>
> Thank you for this insightful comment, which highlights the need to better articulate the theoretical foundation of our core premise. Our motivation stems from a fundamental principle in machine learning: highly complex and non-linear functions are inherently more difficult for models to learn accurately from finite data [1]. This is reflected in the well-known bias-variance trade-off, where approximating a complex target requires immense capacity and data, often leading to failure modes in the most challenging sub-components.
>
> We apply this principle to face forgery detection through the lens of physical image formation. The Formula of Phong illumination model provides a clear hierarchy of complexity: specular reflection is objectively more complex than ambient light and direct light due to its dependence on multiple parameters (viewer direction, surface normals, material exponent) and strong non-linearity (exponential term), making specular reflection more difficult for generators to replicate accurately. They must not only replicate texture but also the intricate, physics-grounded relationships governing light interaction. Thus, inaccuracies in replicating this complex phenomenon serve as a generalizable forgery evidence.
>
> * The formula of Phong illumination model:
> $C_i = {Amb} \ast {T}\_i + \langle {n}\_i , {l} \rangle \cdot {Dir} \ast {T}\_i + \langle{r}\_i , {v}\rangle^n \cdot{Dir} \ast {T}\_i$
>
> where $\mathbf{Amb}$ is ambient light, $\mathbf{Dir}$ is direct light, $\mathbf{T}$ is the texture (i.e., skin albedo), $\langle\mathbf{r}_i , \mathbf{v}\rangle^n \cdot\mathbf{Dir}$ is specular reflection, $\langle , \rangle$ denotes the inner product operation, and $\ast$ denotes the element-wise multiplication. $\mathbf{n}_i$ is the vertex normal of the 3D shape, $\mathbf{l}$ is the direct light direction, $\mathbf{r}_i$ is the reflection direction which can be computed by $\mathbf{r}_i = 2\langle\mathbf{n}_i , \mathbf{l}\rangle\mathbf{n}_i - \mathbf{l}$, $\mathbf{v}$ is the viewer direction, and $n$ is the specular exponent of surface material.
>
>
> Our proposed method is designed to explicitly detect these subtle inconsistencies. The superior performance of SRI-Net across diverse datasets provides strong empirical validation for this theoretically grounded approach. While the visual differences in Fig.3 may be subtle, as the artifacts are often embedded in complex high-frequency patterns. The primary purpose of Fig.3 is to demonstrate that a more accurate specular reflection is obtained when the Retinex-based texture is used as a constraint.
>
>
> [1] Pattern recognition and machine learning[M]. Springer, 2006.

---

> ### Author Response · Authors · 2025-11-19
> **Response to Reviewer yPYK (Part3)**
>
> **3. Clarification of '3D Global Environmental Conditions'**
> >Q1: Could you elaborate on what is meant by '3D global environmental conditions' (line 176, Section 3.1). It is vague.
>
> Thank you for raising this point. We agree that this term can be elaborated more clearly.
>
> In our manuscript, we propose 'Illumination represents a particularly complex physical phenomenon, as it is simultaneously governed by 2D local texture statistics, 3D global environmental conditions, and illumination model constraints, making it a promising avenue for investigation'.
>
> '3D global environmental conditions' refers to the **spatially-varying factors external to the face itself** that collectively determine the illumination on the face surface. These conditions are 'global' because they affect the entire face, and '3D' because their influence is defined within the three-dimensional scene context.
>
> These factors can be analytically characterized through the Phong illumination model:
> $C_i = {Amb} \ast {T}\_i + \langle {n}\_i , {l} \rangle \cdot {Dir} \ast {T}\_i + \langle{r}\_i , {v}\rangle^n \cdot{Dir} \ast {T}\_i$
>
> where $\mathbf{Amb}$ is ambient light, $\mathbf{Dir}$ is direct light, $\mathbf{T}$ is the texture (i.e., skin albedo), $\langle\mathbf{r}_i , \mathbf{v}\rangle^n \cdot\mathbf{Dir}$ is specular reflection, $\langle , \rangle$ denotes the inner product operation, and $\ast$ denotes the element-wise multiplication. $\mathbf{n}_i$ is the vertex normal of the 3D shape, $\mathbf{l}$ is the direct light direction, $\mathbf{r}_i$ is the reflection direction which can be computed by $\mathbf{r}_i = 2\langle\mathbf{n}_i , \mathbf{l}\rangle\mathbf{n}_i - \mathbf{l}$, $\mathbf{v}$ is the viewer direction, and $n$ is the specular exponent of surface material.
>
> As evident from the formulation, accurate illumination estimation requires not only the intensities of ambient and direct light, but also precise knowledge of the direct light direction, surface normal, reflection direction, and viewer direction. The direct light direction is contingent upon the light source position and diffuse conditions, while the viewer direction is determined by the spatial configuration between the viewer and the face.
>
> Crucially, both the direct light (governed by light direction $\mathbf{l}$) and the specular reflection (governed by viewer direction $\mathbf{v}$) are environmental factors external to the face. Since they collectively determine the illumination across the entire face within the 3D scene, we conclude that the illumination is indeed governed by these 3D global environmental conditions.
>
> **4. Clarification of Notation in Formula (1) and Formula (2)**
> >Q2: In Section 3.1 (line 185), there is a potential source of confusion regarding the notation in Formula (1) and Formula (2). Specifically, Formula (1) treats $C$ and $S$ as distinct variables, yet in Formula (2), $S$ is defined as being composed of elements $C_i$. The variable $C_i$ could easily be misinterpreted as a component of $C$, leading to ambiguity and obscuring the intended relationship between $C$, $S$, and $C_i$.
>
> Thank you for this astute observation. The potential confusion arises from the use of the symbol $C$ in two distinct but related contexts across the formulas. We will clarify the intended meaning and the logical flow between them.
>
> In **Formula (1)**, ${I}\_{syn} = \mathcal{R}({S}, {C})$, the term $C$ represents the complete set of color information (encompassing both texture and illumination) for the entire 3D shape $S$. 3D shape $S$ and vertex colors $C$ are the two primary, independent inputs to the renderer $R$.
>
> In **Formula (2)**, $C_i = {Amb} \ast {T}\_i + \langle {n}\_i , {l} \rangle \cdot {Dir} \ast {T}\_i + \langle{r}\_i , {v}\rangle^n \cdot{Dir} \ast {T}\_i$, we drill down to define how the color for a **single vertex** is computed according to the Phong illumination model. The notation $C_i$ refers specifically to the final RGB color of the **i-th** vertex of the 3D shape $S$. It is not that $S$ is composed of $C_i$, but rather that **for each vertex in the shape $S$, we compute its color $C_i$**.
>
> Specifically, $S$ is a 3D mesh of face with shape [53215, 3], where the first dimension represents the number of vertices in the 3D mesh and the second dimension represents the spatial coordinates [X, Y, Z] of each vertex. $C$ also has shape [53215, 3], with the second dimension representing the RGB color values [R, G, B] for each corresponding vertex in $S$. In Formula 2, $C_i$ denotes the RGB value of the i-th vertex in $S$.
>
> **In summary, $C$ represents the complete set of color information for the entire 3D shape $S$. $C$ is the aggregate of all per-vertex colors $C_i$.**

---

> ### Author Response · Authors · 2025-11-19
> **Response to Reviewer yPYK (Part4)**
>
> **5. Clarification on Illumination and Texture Separation in Section 3.2**
> >Q3: For the title 'Illumination and texture separation' in Section 3.2 (line 205), is the intention to decouple the texture and illumination components that were implicitly combined within the variable $C$ as introduced in Formula(1)?
>
> Your understanding is absolutely correct. As shown in Formula 2, the values of $C$ are jointly determined by illumination and texture, where $C$ is a known quantity obtained by registering the face image with the 3D shape $S$. Our objective is to first obtain an accurate texture, and subsequently estimate ambient light and direct light. We then extract the specular reflection component from the illumination through the residual formulation presented in Formula (12) ${SPR} = ({I} - ({H}\gamma)\_{(1-9)} \cdot {T}\_{msr})/{T}\_{msr}$.
>
> **6. Text Structure in Section 3.2 and Clarification on ${T}\_{id}$**
> >Q4: The text spanning lines 236 to 247 in Section 3.2 should be broken down into smaller, logical paragraphs. The current dense formatting is difficult to read. Regarding Formula (6), is there a specific model or methodology suggested for obtaining the identity transformation matrix $T_{id}$? For example, could a pre-trained model such as ArcFace be employed for this purpose, or is an alternative approach required?
>
>
> Thank you for these valuable suggestions regarding readability and technical clarification.  In the revised manuscript, we have restructured the text from lines 236 to 247 into smaller, logical paragraphs to significantly improve readability and flow.
>
> **Clarification on ${T}\_{id}$ and Methodology:**
>
> Formula 6: $T = \overline {T}\_{bfm}  + \mathbf{B}\beta + T\_{id}$
>
> Regarding Formula (6), we wish to clarify that ${T}\_{id}$ is **not a transformation matrix**, but rather the **residual fine-grained texture detail** that the linear PCA model of the Basel Face Model (BFM) fails to capture. Thus, a pre-trained model like ArcFace is not suitable for obtaining ${T}\_{id}$. The reason is fundamental:
> *   **ArcFace** operates on 2D images and produces a high-level, identity-discriminative feature vector in a deep feature space. It is not designed to output a per-vertex, albedo-texture map for a 3D mesh.
> *   **Our ${T}\_{id}$** represents the missing skin texture details for the 3D geometry. The core issue is that the BFM's linear PCA model $\overline{{T}}\_{bfm} + {B}\beta$ is inherently limited. The presence of this unmodeled residual ${T}\_{id}$ leads to inaccurate texture estimation, which subsequently corrupts the precision of the extracted specular reflection.
>
> Therefore, our proposed solution is not to find a feature vector for ${T}\_{id}$, but to **replace the BFM texture model entirely with Retinex-based method**. This approach directly estimates a more accurate and detailed texture map $ {T}\_{msr} $, effectively minimizing the residual ${T}\_{id}$ by design, which in turn enables more precise specular reflection extraction and accelerates the process.

---

> ### Comment · Reviewer_yPYK · 2025-11-25
>
> I thank the authors for their detailed response and for revising the manuscript. However, after carefully revisiting the paper and the rebuttal, I maintain significant concerns regarding the method's novelty and theoretical grounding.
>
> My primary concern lies in the distinction between this work and prior art (Chen et al., 2022). The rebuttal differentiates the method by claiming *"Multi-Scale Retinex (MSR) serves not as an input modality."* However, this statement is directly contradicted by **Figure 2**, which clearly shows the Retinex-based `Texture` being warped and fed into **`Entry_flow_1`** as a primary input feature ($f_{tex}$). While I acknowledge that MSR is used to compute the specular reflection component, the fact that the MSR texture *also* serves as a direct input branch structurally aligns this method with the "MSR stream" design in Chen et al. (2022). Therefore, the claim that Retinex functions *solely* as a constraint is factually incorrect based on the network design.
>
> Regarding the theoretical foundation and the clarification on "3D global environmental conditions," I remain unconvinced. In the rebuttal, the authors emphasize that illumination is governed by these 3D conditions (light/view directions) to justify the model. **However, this argument overlooks the critical gap between the physical formula and the actual simulation process.** Validating the Phong model's formula is not the same as validating the *inverse rendering* from a single 2D image. The authors ignore the inherent ambiguity and potential information loss in this simulation. Without the requested statistical evidence (e.g., feature distribution analysis), there is no proof that the extracted "specular reflection" captures genuine forgery traces rather than merely reflecting the inevitable artifacts of this imperfect 3D simulation.
>
> Consequently, the over-packaging of terminology and the lack of statistical backing for the physics-based claims prevent me from raising my score. I look forward to any further clarification.

---

> > ### Author Response · Authors · 2025-11-26
> > **Response to Reviewer yPYK (R1,Part1)**
> >
> > Thank you for your valuable feedback and constructive suggestions, which have been instrumental in improving our paper. The responses to the concerns are as follows. We have revised the manuscript accordingly, and the changes have been highlighted in red for your convenience.
> >
> > >R1W1: Regarding the theoretical foundation and the clarification on "3D global environmental conditions," I remain unconvinced. In the rebuttal, the authors emphasize that illumination is governed by these 3D conditions (light/view directions) to justify the model. However, this argument overlooks the critical gap between the physical formula and the actual simulation process. Validating the Phong model's formula is not the same as validating the inverse rendering from a single 2D image. The authors ignore the inherent ambiguity and potential information loss in this simulation. Without the requested statistical evidence (e.g., feature distribution analysis), there is no proof that the extracted "specular reflection" captures genuine forgery traces rather than merely reflecting the inevitable artifacts of this imperfect 3D simulation.
> >
> > We completely understand and agree with your point that "due to the inherent ambiguity and potential information loss in the inverse rendering from a single 2D image (which primarily exists in the process of components decomposition and flattening them into UV space), statistical evidence is needed to prove that the extracted 'specular reflection' captures genuine forgery traces rather than merely reflecting the inevitable artifacts of this imperfect 3D simulation."
> >
> > In our manuscript, the first part of the ablation studies (Sec. 4.4, "Effectiveness of each branch") provides relevant statistical evidence. Rows (1) Img and (2) SPR represent the metrics obtained when solely using the flattened original face image or the flattened specular reflection as input to XceptionNet, trained on the FF++ dataset and tested on the Celeb_v2, DF40, and DiFF datasets.
> >
> > The underlying logic is this: **If the specular reflection component were merely reflecting the inevitable artifacts of the imperfect 3D simulation (a phenomenon that should occur in both real and fake faces), then using it alone as network input should not enable effective real/fake classification, and its performance should be close to random guessing (i.e., AUC ~50%).**
> >
> > **However, as the results show, using specular reflection as the sole input (i.e., (2) SPR) already achieves considerable AUC performance. This evidence strongly indicates that the specular reflection component indeed captures genuine forgery traces from the face image, which are discriminative and generalize across datasets.**
> >
> > To provide more comprehensive statistical evidence, we performed both **PCA visualization and score distribution analysis** on the features extracted when using specular reflection as the sole input to XceptionNet, evaluated on the DiFF dataset. The PCA visualization, which projects the original 2048-dimensional feature vectors onto a 2D space, does not clearly reveal distinct distribution patterns between real and fake faces. This limited visual separability is expected given the model's AUC of 73.9% and the substantial information loss inherent in reducing high-dimensional features to just two dimensions. However, our score distribution histogram demonstrates a statistically significant divergence between real and fake samples. The clear separation in score distributions provides compelling evidence that the specular reflection component captures discriminative forgery traces capable of effectively distinguishing between real and fake faces. The corresponding visualizations have been included in Appendix A.6 of the revised manuscript.
> >
> > We acknowledge that the initial version of our manuscript did not provide a sufficiently detailed explanation of the implications of each row in the ablation study table. We have revised the relevant sections in the revised manuscript. We also wish to express our sincere gratitude for your comment on the necessity to distinguish genuine forgery traces in the specular reflection from potential simulation artifacts. This feedback was crucial for enhancing the overall rigor and validity of our study.
> >
> > **Ablation study on Effectiveness of each branch of SRI-Net. (AUC(%))**
> >
> > | Method Settings       | Celeb_v2 | DF40 | DiFF |
> > |-----------------------|----------|------|------|
> > | (1) Img               | 72.1     | 74.2 | 66.0 |
> > | (2) SPR               | 75.7     | 78.8 | 73.9 |
> > | (3) Img + SPR         | 84.5     | 87.2 | 83.9 |
> > | (4) Img + SPR + MSR/Dir | 87.5   | 90.9 | 86.8 |

---

> > ### Author Response · Authors · 2025-11-26
> > **Response to Reviewer yPYK (R1,Part2)**
> >
> > >R1W2: My primary concern lies in the distinction between this work and prior art (Chen et al., 2022). The rebuttal differentiates the method by claiming "Multi-Scale Retinex (MSR) serves not as an input modality." However, this statement is directly contradicted by Figure 2, which clearly shows the Retinex-based Texture being warped and fed into Entry_flow_1 as a primary input feature. While I acknowledge that MSR is used to compute the specular reflection component, the fact that the MSR texture also serves as a direct input branch structurally aligns this method with the "MSR stream" design in Chen et al. (2022). Therefore, the claim that Retinex functions solely as a constraint is factually incorrect based on the network design.
> >
> > Thank you for this clarification. We acknowledge the oversight in our previous response regarding the role of the Retinex-based texture and apologize for any confusion caused by the statement "Retinex functions solely as a constraint.". In our manuscript, the role of MSR is primarily reflected in two aspects:
> >
> > **Role A**: As a critical intermediate module that facilitates more accurate specular reflection extraction (As we described in response to W1).
> >
> > **Role B**: It is true that the MSR-based texture serves as an input to our network. However, its functional role in our framework is fundamentally different from that in Chen et al. (2022).
> >
> > In Chen et al. (2022), the MSR component is employed as a supplemental modality designed to provide illumination-invariant features, which are then fused with the original image features to enhance discrimination.
> >
> > In contrast, within our method, its core role is to collaborate with the specular reflection (SPR) and direct light (Dir) components to collectively enable the extraction of the physically grounded concept we define as 'specular reflection inconsistency'. The definition of 'specular reflection inconsistency' is rooted in the formula of the Phong illumination model:
> >
> > $C_i = \mathbf{Amb} \ast \mathbf{T}_i + \langle \mathbf{n}_i, \mathbf{l} \rangle \cdot \mathbf{Dir} \ast \mathbf{T}_i + \langle \mathbf{r}_i, \mathbf{v} \rangle^n \cdot \mathbf{Dir} \ast \mathbf{T}_i$.
> >
> > where $\mathbf{Amb}$ is ambient light, $\mathbf{Dir}$ is direct light, $\mathbf{T}$ is the texture (i.e., skin albedo), $\langle\mathbf{r}_i , \mathbf{v}\rangle^n \cdot\mathbf{Dir}$ is specular reflection, $\langle , \rangle$ denotes the inner product operation, and $\ast$ denotes the element-wise multiplication. $\mathbf{n}_i$ is the vertex normal of the 3D shape, $\mathbf{l}$ is the direct light direction, $\mathbf{r}_i$ is the reflection direction which can be computed by $\mathbf{r}_i = 2\langle\mathbf{n}_i , \mathbf{l}\rangle\mathbf{n}_i - \mathbf{l}$, $\mathbf{v}$ is the viewer direction, and $n$ is the specular exponent of surface material.
> >
> > The specular reflection component, $\mathbf{SPR} = \langle \mathbf{r}_i, \mathbf{v} \rangle^n \cdot \mathbf{Dir}$, establishes a precise physical constraint: for a given direct light direction $\mathbf{l}$, 3D shape vertex normal $\mathbf{n}_i$, viewer direction $\mathbf{v}$, direct light intensity $\mathbf{Dir}$, and face texture $\mathbf{T}_i$ (Given that human faces share similar skin properties, the relative material characteristics $n$ can be approximated through face texture.), the specular reflection must follow a specific, deterministic pattern in a real face.
> >
> > In our pipeline, we first flatten the specular reflection, face texture, and direct light components into UV space to normalize their directions (norm direct light direction $\mathbf{l}$, 3D shape vertex normal $\mathbf{n}_i$, viewer direction $\mathbf{v}$).  In a real face, under the constraints of the Phong illumination model, the specular reflection exhibits a specific and consistent pattern relative to the corresponding face texture and direct light. However, the specular reflection component is objectively more complex and difficult for generative models to replicate perfectly due to its dependence on multiple parameters and strong non-linearity. Therefore, in forged images, the specular reflection will deviate from the pattern characteristic of a real face given the corresponding texture and direct light. This deviation is the "specular reflection inconsistency" we defined, a breakdown in the multi-attribute relational constraint imposed by the Phong illumination model.

---

> ### Author Response · Authors · 2025-11-26
> **Response to Reviewer yPYK (R1,Part3)**
>
> Thus, the core of SRI-Net is not to learn an arbitrary pattern from the fusion of input components but to learn how these components should consistently relate under the Phong illumination model's constraints.
>
> *   The first cross-attention stage (Formula 13) learns the baseline relationship between texture and direct light ($f_{td}$).
> *   The second stage (Formula 14) then evaluates how well the actual specular reflection aligns with this expected baseline. The output feature $f_{std}$ is the learned embedding that encodes the degree and nature of the multi-attribute inconsistency under the Phong illumination model.
>
> To provide statistical support for this theoretical analysis, we have incorporated an additional ablation experiment in Sec. 4.4, where the MSR texture is used as the sole input to the XceptionNet (i.e., row (3) 'MSR'). The results demonstrate that using MSR alone yields suboptimal performance. However, a marked improvement is observed when MSR is utilized in conjunction with the specular reflection (SPR) and direct light (Dir) components (compare rows (4) and (5)). This contrast substantiates that the primary role of the MSR texture in our framework is not as a standalone feature, but as a key element in extracting the physically meaningful 'specular reflection inconsistency'.
>
>
>
> | Method Settings       | Celeb_v2 | DF40 | DiFF |
> |-----------------------|----------|------|------|
> | (1) Img               | 72.1     | 74.2 | 66.0 |
> | (2) SPR               | 75.7     | 78.8 | 73.9 |
> | **(3) MSR (new added)**              | 68.5    | 71.0 | 60.8 |
> | (4) Img + SPR         | 84.5     | 87.2 | 83.9 |
> | (5) Img + SPR + MSR/Dir | 87.5   | 90.9 | 86.8 |

---

### Official Review · Reviewer_NiAP · 2025-10-31

**Soundness:** 3
**Presentation:** 3
**Contribution:** 2
**Rating:** 4
**Confidence:** 5

**Summary:**

This paper presents a physics-based face forgery detection method that exploits the difficulty of reproducing specular reflection in generative models. The authors propose a Retinex-based texture estimation for clean illumination decomposition, a residual spherical-harmonics model for efficient specular separation, and an SRI-Net that learns physical correlations among reflection, light, and texture.

**Strengths:**

- This manuscript is well-structured and easy to follow.
- This work provides a clear physical motivation by identifying specular reflection as the most complex component in generated faces, which is a reasonable choice grounded in physical principles.

**Weaknesses:**

- Retinex-based reflectance estimation has already been used in face forensics and related vision tasks[1,2] for exposing subtle forgery cues, so the usage of Multi-Scale Retinex is not entirely novel.
- This work does not sufficiently validate robustness to common post-processing, such as Gaussian Blur and JPEG compression.

[1] Attention-based Two-stream Convolutional Networks for Face Spoofing Detection. IEEE TIFS 2020.

[2] Exposing Face Forgery Clues via Retinex-based Image Enhancement. ACCV 2022.

**Questions:**

Specular reflection depends on surface properties. How stable is this method on faces with different skin tones (e.g., using AI-Face [3])?


[3] AI-Face: A Million-Scale Demographically Annotated AI-Generated Face Dataset and Fairness Benchmark. CVPR 2025.

**Details Of Ethics Concerns:**

No ethical concerns identified.

---

> ### Author Response · Authors · 2025-11-19
> **Response to Reviewer NiAP (Part1)**
>
> Thank you for your thoughtful feedback and valuable suggestions, which helped us a lot to make the paper better. The responses to the concerns are as follows. We have revised the manuscript accordingly, and the changes have been highlighted in blue for your convenience.
>
> **1. The usage of Multi-Scale Retinex**
>
> >W1: Retinex-based reflectance estimation has already been used in face forensics and related vision tasks[1,2] for exposing subtle forgery cues, so the usage of Multi-Scale Retinex is not entirely novel.
> [1] Attention-based Two-stream Convolutional Networks for Face Spoofing Detection. IEEE TIFS 2020.
> [2] Exposing Face Forgery Clues via Retinex-based Image Enhancement. ACCV 2022.
>
> Thanks for your comment and for pointing out these relevant works. We agree that Retinex theory has been previously applied in face forensics tasks [1,2], and we have cited these papers in the revised manuscript to better situate our contribution. In our manuscript, the novelty lies not in the use of Retinex-based texture estimation, but in its novel role within our designed physics-driven framework for exposing forgery evidence.
>
> The objective of [1,2] is to exploit Retinex-based image enhancement to reveal subtle spatial domain artifacts in forged regions. In these approaches, Retinex representations serve as a new modality that is directly fed into the network as input.
>
> In contrast, our objective centers on physics-based illumination decomposition. Motivated by the Phong illumination model formulation, we identify specular reflection as a robust forgery indicator due to its complex multi-parameter formulation, stronger nonlinearity and inherent difficulty to replicate accurately. Our goal is to precisely isolate and extract the specular component, subsequently exploiting inconsistencies between specular reflection, face texture and direct light to capture forgery evidence. This represents a physics-grounded analytical framework rather than mere image enhancement.
>
> In our pipeline, Multi-Scale Retinex (MSR) serves not as an input modality but rather as a critical intermediate module that facilitates more accurate specular reflection extraction. As detailed in Section 3.2, the widely used Basel Face Model (BFM) fails to capture fine-grained facial textures, leading to contaminated specular reflection estimates. Our MSR-based texture $T_{msr}$ is specifically designed to replace and outperform the BFM texture model. This allows for a purer separation of the specular component via our proposed residual-based approach (Formula 12: ${SPR} = ({I} - ({H}\gamma)\_{(1-9)} \cdot {T}\_{msr})/{T}\_{msr}$).
>
> In general, while we build upon the established Retinex theory, its application in our work is contextually and functionally distinct. Previous works use it for general enhancement; we use it as a precision tool for 3D illumination decomposition to estimate more accurate specular reflection to capture more robust forgery evidence.
>
> **2. Robustness to common post-processing**
> >W2: This work does not sufficiently validate robustness to common post-processing, such as Gaussian Blur and JPEG compression.
>
> Thank you for this valuable suggestion. We agree that robustness to common post-processing operations is a critical requirement for a practical forgery detection detector. Following the advice, we have conducted extensive new experiments on Celeb_v2, DF40 and DiFF to evaluate the robustness of our proposed SRI-Net under three types of common post-processing on test images: **Gaussian Blurring**, **JPEG Compression**, and **Gaussian Noise**.
>
> The results show that SRI-Net has demonstrated good stability, maintaining a high AUC performance. We attribute this robustness to the physics-driven nature of SRI-Net. While post-processing operations can degrade high-frequency forgery evidence that many spatial-domain methods rely on, the physical inconsistencies in specular reflection, direct light and texture that SRI-Net is designed to capture are more deeply embedded in the image formation process. These inconsistencies manifest as violations of physical laws in the illumination domain, which are more resilient to image distortions like blurring and compression.
>
> **AUC (%) performance comparison under Common Post-processing.**
>
> | Perturbation       | Celeb_v2 | DF40 | DiFF |
> | :----------- | :------- | :-------------------- | :---------------------- |
> | Original Image       | 87.5     | 90.9                  | 86.8               |
> | Gaussian Blurring    | 86.6     | 88.7                  | 84.2               |
> | JPEG Compression   | 87.1     | 90.2                  | 86.4                    |
> | Gaussian Noise |   87.0   |     88.9         |         85.5            |

---

> ### Author Response · Authors · 2025-11-19
> **Response to Reviewer NiAP (Part2)**
>
> **3. Robustness across skin tones**
> >Q1: Specular reflection depends on surface properties. How stable is this method on faces with different skin tones (e.g., using AI-Face [3])?
> [3] AI-Face: A Million-Scale Demographically Annotated AI-Generated Face Dataset and Fairness Benchmark. CVPR 2025.
>
> We appreciate this critical question regarding robustness across skin tones. To address it directly, we have conducted a new evaluation using the AI-Face dataset. The results in Tab. 6 of the revised manuscript demonstrate that our method achieves superior and consistent performance across all three skin tone groups. This provides empirical evidence of its robustness.
>
> **AUC(%) performance comparison on AI-Face dataset.**
>
> | Method                      | Light | Medium | Dark |
> | :-------------------------- | :---- | :----- | :--- |
> | Xception                | 97.69 | 98.44  | 98.88 |
> | EffiNet-B4              | **99.23** | 98.94  | 97.59 |
> | F3-Net                  | 98.51 | 98.68  | 98.79 |
> | CORE                    | 97.80 | 98.47  | 98.79 |
> | DAG-FDD                 | 97.56 | 98.79  | 98.73 |
> | SRI-Net (Ours)          | 99.12 | **99.01** | **99.18** |
>
>
> The observed stability can be explained through the formula of the Phong illumination model.
>
> $C_i = {Amb} \ast {T}\_i + \langle {n}\_i , {l} \rangle \cdot {Dir} \ast {T}\_i + \langle{r}\_i , {v}\rangle^n \cdot{Dir} \ast {T}\_i$
>
> In this model, skin tone variations primarily reside in the ambient light component (as shown in Fig.4 of the manuscript), which interacts uniformly with the skin's diffuse albedo. A pivotal strength of our pipeline lies in its explicit separation of illumination. Our Retinex-based texture extraction, followed by spherical harmonic modeling, collaboratively works to isolate and filter out this ambient light component. By effectively removing the primary carrier of skin tone information, our method becomes largely invariant to such variations. While specular reflection is indeed a surface property, our formulation shows it is primarily linked to fine-scale textural detail rather than the macro-scale skin tone, which is modeled as an illumination effect. This design inherently decouples the detection from skin tone, leading to robust performance across demographics.

---

> ### Comment · Area_Chair_bUzo · 2025-11-28
> **Rebuttal Review Request**
>
> Dear Reviewers,
>
> Thank you for your time and thoughtful feedback on this manuscript.
>
> The authors have now submitted their rebuttal. If you haven’t already, we kindly ask you to review their responses and consider whether your concerns have been adequately addressed.
>
> Best regards,
>
> AC

---

### Author Response · Authors · 2025-12-01
**General Response**

Dear AC and Reviewers,

Thank you for handling our manuscript and for the constructive feedback from the reviewers. We would like to briefly follow up with this general response, clarifying our main contributions and summarizing how we have addressed common concerns.

**>Main Contributions**

**1. New forgery detection perspective:** We propose specular reflection is a robust and generalizable forgery indicator, based on its complex and non-linear physical formulation within the Phong illumination model and inherent difficulty for generative models to accurately replicate.

**2. Improved texture estimation method:** We introduce a Retinex-based texture extraction approach, enabling faster and more accurate separation of illumination and texture for precise specular reflection extraction.

**3. Cross-attention based inconsistency modeling:** Our proposed SRI-Net incorporates a two-stage cross-attention mechanism to explicitly model the inconsistencies among specular reflection, face texture, and direct light, thereby extracting forgery evidence grounded in physical illumination constraints.

**4. Superior detection performance:** Extensive experiments demonstrate that SRI-Net achieves state-of-the-art results on both traditional and generative deepfake datasets, showcasing strong generalization and robustness.

**>Common Concerns and Clarification**

**1. Theoretical foundation of ''Specular reflection is harder to replicate'':** Our method is grounded in the machine learning principle that highly complex and non-linear functions are inherently more difficult for models to learn accurately from finite data. This is reflected in the Phong illumination model, where specular reflection depends on more parameters and exhibits stronger nonlinearity than other components, making it particularly challenging to replicate and thus a reliable source of generalizable forgery evidence.

**2. The usage of Retinex:** We acknowledge that Retinex-based texture estimation has been used in face forensics works as an independent illumination-invariant modal. Our work differs fundamentally from such approaches by applying Retinex texture as a critical component within a physics-driven illumination decomposition pipeline, enabling more precise extraction of specular reflection and the extraction of specular reflection inconsistency under Phong illumination model constraints.

**3. Mitigation Strategies for Failure Cases:** We have expanded Appendix A.5 to address failures under extreme poses and occlusions, outlining two concrete improvement strategies: using pixel-accurate 2D landmarks (e.g., Pixel-in-Pixel Net) to correct 3D shape misalignments, and adopting more advanced 3D shape estimation (e.g., 3DDFA-V3) for more robust shape estimation.

---

### Meta-Review · Area_Chair_bpDW · 2026-01-07

**Summary:**

This paper proposes SRI-Net for face forgery detection, motivated by the hypothesis that specular reflection (under a Phong-style illumination model) is harder for modern generators to replicate consistently. The method uses Retinex-based texture estimation to support illumination decomposition / specular separation, then a two-stage cross-attention module to model inconsistencies among specular reflection, texture, and direct light.

Reviewers found the paper generally clear and the cross-dataset results strong, but raised concerns about (i) novelty vs. prior Retinex-stream forensics, (ii) whether the “physics grounding” is fully validated given single-image inverse rendering ambiguity, and (iii) robustness limits due to dependence on 3D fitting.

The paper presents a plausible physics-motivated cue and demonstrates strong cross-dataset performance; the rebuttal further strengthens the work with added post-processing robustness and demographic (skin-tone) evaluations, so I recommend acceptance (poster).

**Reviewer Concerns:**

Addressed by rebuttal:

1. Robustness checks: Added experiments under common post-processing (blur/JPEG/noise) showing relatively stable AUC.

2. Demographic robustness: Added AI-Face evaluation with skin-tone breakdown reporting consistent performance.

3. Clarifications & ablations: Improved notation/terminology, provided more discussion of failure cases and mitigations, and added ablations supporting design choices (e.g., attention vs. alternatives; MSR-only vs. combined inputs).

4. Related work: Cited relevant Retinex-based prior works and clarified the role of Retinex within the pipeline (including acknowledging an earlier overstatement).

Still outstanding:

1. Novelty remains contested: A reviewer argues the Retinex texture is clearly used as a direct input branch, making the distinction from prior Retinex-stream designs less sharp; overall novelty is still somewhat incremental.

2. Physics/validity gap: While Phong-model motivation is plausible, it is still not fully established that the extracted “specular reflection” consistently captures forgery traces rather than artifacts introduced by imperfect single-image inverse rendering / UV flattening. Added analyses help, but do not fully resolve this conceptual concern.

**Reviewer Scores:**

NiAP: 4 → 4 (no post-rebuttal update provided)

yPYK: 4 → 4 (explicitly maintains concerns; no score increase indicated)

KKiD: 6 → 6 (explicitly: “maintain my original rating”)

TtYa: 6 → 6 (raised a follow-up concern; no score change stated)

---

### Decision · Program_Chairs · 2026-01-26

Accept (Poster)